# Two complement receptor one alleles have opposing associations with cerebral malaria and interact with α+thalassaemia

D Herbert Opi[1,2‡§], Olivia Swann[2‡*], Alexander Macharia[1], Sophie Uyoga[1], Gavin Band[3], Carolyne M Ndila[1], Ewen M Harrison[4], Mahamadou A Thera[5], Abdoulaye K Kone[5], Dapa A Diallo[5], Ogobara K Doumbo[5], Kirsten E Lyke[6], Christopher V Plowe[6], Joann M Moulds[7], Mohammed Shebbe[1], Neema Mturi[1], Norbert Peshu[1], Kathryn Maitland[1,8], Ahmed Raza[2], Dominic P Kwiatkowski[3,9], Kirk A Rockett[3], Thomas N Williams[1,8†], J Alexandra Rowe[2†]

[1]Kenya Medical Research Institute-Wellcome Trust Research Programme, Kilifi, Kenya; [2]Centre for Immunity, Infection and Evolution, Institute of Immunology and Infection Research, School of Biological Sciences, University of Edinburgh, Edinburgh, United Kingdom; [3]Wellcome Trust Centre for Human Genetics, University of Oxford, Oxford, United Kingdom; [4]Centre for Medical Infomatics, Usher Insitute of Population Health Sciences and Informatics, University of Edinburgh, Edinburgh, United Kingdom; [5]Malaria Research and Training Centre, Faculty of Medicine, Pharmacy, and Dentistry, University of Bamako, Bamako, Mali; [6]Division of Malaria Research, Institute for Global Health, University of Maryland School of Medicine, Baltimore, United States; [7]Lifeshare Blood Centers, Shreveport, United States; [8]Department of Medicine, Imperial College, London, United Kingdom; [9]Wellcome Trust Sanger Institute, Cambridge, United Kingdom

*For correspondence:
Olivia.Swann@ed.ac.uk

[†]These authors also contributed equally to this work
[‡]These authors also contributed equally to this work

Present address: [§]Burnet Insititue for Medical Research and Public Health, Melbourne, Australia

Competing interests: The authors declare that no competing interests exist.

**Abstract** Malaria has been a major driving force in the evolution of the human genome. In sub-Saharan African populations, two neighbouring polymorphisms in the Complement Receptor One (*CR1*) gene, named *Sl2* and *McC^b*, occur at high frequencies, consistent with selection by malaria. Previous studies have been inconclusive. Using a large case-control study of severe malaria in Kenyan children and statistical models adjusted for confounders, we estimate the relationship between *Sl2* and *McC^b* and malaria phenotypes, and find they have opposing associations. The *Sl2* polymorphism is associated with markedly reduced odds of cerebral malaria and death, while the *McC^b* polymorphism is associated with increased odds of cerebral malaria. We also identify an apparent interaction between *Sl2* and α+thalassaemia, with the protective association of *Sl2* greatest in children with normal α-globin. The complex relationship between these three mutations may explain previous conflicting findings, highlighting the importance of considering genetic interactions in disease-association studies.
DOI: https://doi.org/10.7554/eLife.31579.001

## Introduction

Complement Receptor One (CR1) plays a key role in the control of complement activation and the immune clearance of C3b/C4b-coated immune complexes (*Krych-Goldberg and Atkinson, 2001*). CR1 is expressed on a range of cells including red blood cells (RBCs), leucocytes and glomerular podocytes (*Krych-Goldberg and Atkinson, 2001*). A number of CR1 polymorphisms have been described, including four molecular weight variants and variation in the number of CR1 molecules

**eLife digest** Malaria kills more than half a million children in Africa every year. The disease is caused by the *Plasmodium falciparum* parasite, and mosquitos infected with the parasites spread them to humans when they bite. Once inside a human, the parasites infect the red blood cells. In severe cases, these red blood cells can stick to the walls of small blood vessels that supply the brain and so hinder the flow of oxygen, causing a coma. This is called cerebral malaria. Malaria can also result in the destruction of many oxygen-carrying red blood cells, which causes severe anemia. Both cerebral malaria and severe anemia can lead to death.

Small changes (called mutations) in certain human genes can protect against malaria. Over time, mutations that protect people living in Africa from dying from malaria have been passed down through generations. A good example is the sickle cell mutation, which causes red blood cells to be of an unusual shape, but also affects the ability of malaria parasites to grow normally within red cells. Finding new mutations that protect against malaria may help scientists understand how severe malaria happens and eventually develop new drugs and vaccines against the disease. Some studies have found that mutations in a gene called complement receptor 1 (CR1) may be protective, although others have disagreed.

Now, Opi, Swann et al. show that children with one of the CR1 mutations were one-third less likely to get cerebral malaria and half as likely to die as children without the mutation. In the study, genetic and health information on more than 5,500 children in Kenya were analyzed to see if the severity of malaria differed depending on whether they had a CR1 mutation. They also found that the CR1 mutation is only protective against severe malaria when the child does not have another malaria- protective mutation called α-thalassemia. In children with α-thalassemia, the CR1 mutation does not make a difference.

The interaction between the CR1 mutation and α-thalassemia may explain why some studies did not show a benefit of CR1. If the researchers did not include α-thalassemia in their assessment, they could not have seen the whole picture. Future studies showing how the CR1 mutation protects against cerebral malaria could help identify new treatments that prevent severe disease or death. More study of interactions between genes that play a role in malaria may also be helpful.

DOI: https://doi.org/10.7554/eLife.31579.002

expressed on the surface of RBCs (reviewed by [*Krych-Goldberg et al., 2002*; *Schmidt et al., 2015*]). Missense mutations of CR1 form the basis of the Knops blood group system of antigens, that includes the antithetical antigen pairs of Swain-Langley 1 and 2 (Sl1 and Sl2) and McCoy a and b (McC[a] and McC[b]) (*Moulds, 2010*). The non-synonymous single nucleotide polymorphisms (SNPs) A4828G (rs17047661) and A4795G (rs17047660) within exon 29 of the *CR1* gene give rise to the *Sl1/*

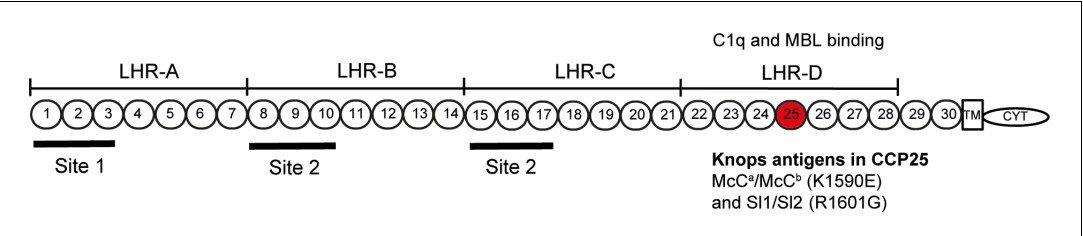

**Figure 1.** Diagram of the most common Complement Receptor 1 size variant (CR1*1). Adapted from *Schmidt et al. (2015)* and *Krych-Goldberg et al. (2002)*. The ectodomain of CR1 is composed of 30 Complement Control Protein (CCP) domains which are organized into four 'Long Homologous Repeats' (LHR). The single-nucleotide polymorphisms determining the Sl and McC antigens of the Knops blood group system are found in CCP 25 in LHR-D (red). Various functions have been mapped to different regions of CR1, including Site 1 (decay accelerating activity for C3 convertases; binding of the complement component C4b and the *P. falciparum* invasion ligand PfRH4), and Site 2 (cofactor activity for Factor I; binding of C3b and C4b and *P. falciparum* rosetting). LHR-D is thought to bind C1q and Mannose Binding lectin (MBL), but the specific binding sites have not been mapped. TM, transmembrane region; CYT, cytoplasmic tail.

DOI: https://doi.org/10.7554/eLife.31579.003

*Sl2* and *McC^a/McC^b* alleles, encoding R1601G and K1590E, respectively (*Moulds et al., 2001*) (*Figure 1*).

CR1 has been implicated in the pathogenesis of multiple diseases, with epidemiological and in vitro data suggesting a role in malaria (*Schmidt et al., 2015*). The *Sl2* and *McC^b* alleles occur at high frequencies only in populations of African origin (*Figure 2*) (*Thathy et al., 2005*; *Zimmerman et al., 2003*; *Moulds et al., 2004*; *Noumsi et al., 2011*; *Fitness et al., 2004*; *Covas et al., 2007*; *Gandhi et al., 2009*; *Yoon et al., 2013*; *Hansson et al., 2013*; *Kariuki et al., 2013*; *Eid et al., 2010*), which, given the historical prevalence of the malaria-causing parasite *Plasmodium falciparum* in sub-Saharan Africa, might suggest a possible survival advantage against malaria (*Rowe et al., 1997*; *Rowe et al., 2000*). CR1 is a receptor for the invasion of RBCs by *Plasmodium falciparum* merozoites (*Spadafora et al., 2010*; *Tham et al., 2010*) and for the formation of clusters of *P. falciparum*-infected RBCs (iRBCs) and uninfected RBCs, known as rosettes (*Rowe et al., 1997*). The rosetting phenotype is associated with severe malaria in sub-Saharan Africa (*Doumbo et al., 2009*), with pathological effects likely due to the obstruction of microcirculatory blood flow (*Kaul et al., 1991*). RBCs from donors with the high-frequency African CR1 Knops mutations bind poorly to the parasite ligand *P. falciparum* erythrocyte membrane protein-1 (PfEMP1) that mediates rosetting by iRBCs, potentially protecting against severe malaria by reducing rosetting (*Rowe et al., 1997*). Nevertheless, epidemiological data supporting this possibility are contradictory, with some studies showing an association between *Sl* and *McC* genotypes and severe malaria (*Thathy et al., 2005*; *Kariuki et al., 2013*; *Tettey et al., 2015*) and others finding none (*Zimmerman et al., 2003*; *Hansson et al., 2013*; *Jallow et al., 2009*; *Manjurano et al., 2012*; *Toure et al., 2012*; *Rockett et al., 2014*). Some previous studies have not considered *Sl* and *McC* genotypes together in the same statistical model, despite their physical adjacency in the CR1 molecule, nor taken into account potential interactions with other malaria resistance genes. Given the important biological role of CR1 in malaria host-parasite interactions, we aimed to clarify the relationship between the *Sl* and *McC* alleles and severe malaria in a case-control study of Kenyan children. These investigations were supplemented with a separate longitudinal cohort study of Kenyan children, examining the associations of these alleles with uncomplicated malaria and other common childhood illnesses. Finally, we also investigated the influence of these alleles on the formation of *P. falciparum* rosettes, as a potential functional explanation for these results through ex vivo laboratory studies.

## Results

### The *Sl2/Sl2* genotype is associated with protection against cerebral malaria and death in the Kenyan case-control study

Data were obtained from 5545 children enrolled in a case-control study of severe malaria (*Figure 3*). The general characteristics of the cases and controls are shown in *Supplementary file 1A*, and the characteristics of the dataset by *Sl* and *McC* genotype are shown in *Supplementary file 1B*. The *Sl2* and *McC^b* allele frequencies (0.68 and 0.16 respectively) were comparable to other African populations (*Figure 2*). There was no significant deviation from Hardy-Weinberg equilibrium for the *Sl* or *McC* genotypes among controls (*Supplementary file 1C*).

Using a simple logistic regression model containing only *Sl* and *McC* genotypes (referred to as the unadjusted analysis below), we found a non-significant association between the *Sl2* allele and severe malaria overall, with the *Sl2/Sl2* genotype being associated with an OR for severe malaria of 0.90 (95% CI 0.79–1.01; p=0.07) (*Supplementary file 1D*). We attempted to refine this signal by fitting a more complete model to the data, including the potential confounding factors of ethnicity, location, sickle cell trait, ABO blood group and $\alpha^+$thalassaemia genotype, as well as considering possible first-order interactions between terms (referred to as the full adjusted analysis below). A significant protective association was observed for *Sl2* in the recessive form (adjusted Odds Ratio (aOR) 0.78; 95% CI 0.64–0.95; p=0.011), which was most marked for cerebral malaria (aOR 0.67; 0.52–0.87; p=0.006) (*Figure 4* and *Table 1*). The *Sl2/Sl2* genotype was also associated with significant protection against death from severe malaria (aOR 0.50; 0.30–0.80; p=0.002), and death among children admitted with a specific diagnosis of cerebral malaria in the full adjusted analysis (aOR 0.44; 0.23–0.78; p=0.007) (*Figure 4* and *Table 1*). Unexpectedly, we observed a significant interaction between *Sl2* and $\alpha^+$thalassaemia genotype, such that the protective associations of *Sl2* were only

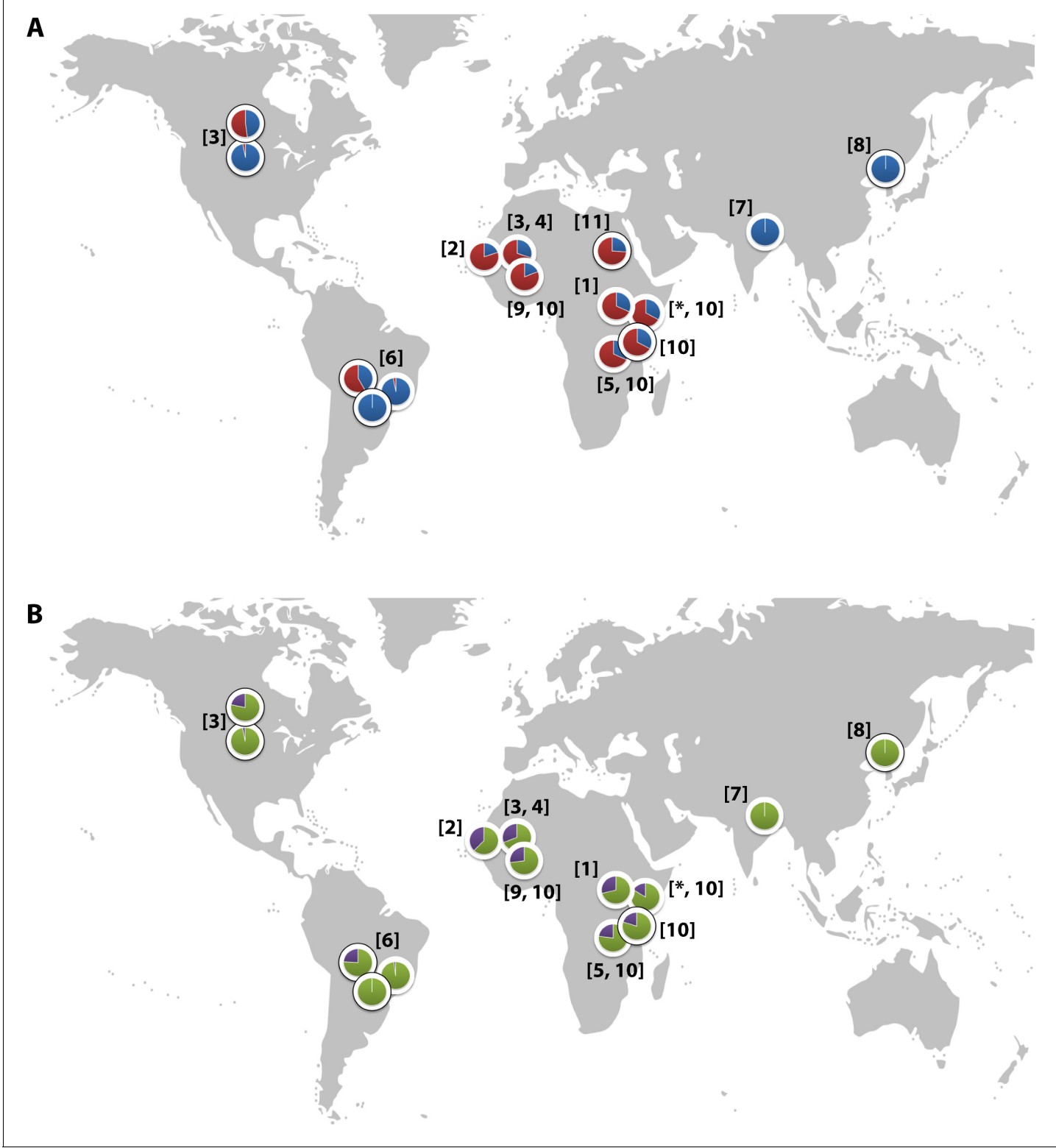

**Figure 2.** Global distribution of the CR1 Knops *Sl* and *McC* alleles  (A) Shows the global frequencies of the *Sl* alleles. *Sl1* is represented in blue and *Sl2* in red. (B) Shows the global frequencies of the *McC* alleles. *McC^a* is represented in green and *McC^b* in purple. The two samples in North and South America showing high frequencies of *Sl2* and *McC^b* alleles are both derived from populations with African heritage. Numbers in parentheses indicate the studies from which the *Sl* and *McC* allele frequencies were derived, with * indicating data derived from this study. [1] *Thathy et al., 2005*; [2] *Figure 2 continued on next page*

*Figure 2 continued*

*Zimmerman et al., 2003*; [3] *Moulds et al., 2004*; [4] *Noumsi et al., 2011*; [5] *Fitness et al., 2004*; [6] *Covas et al., 2007*; [7] *Gandhi et al., 2009*; [8] *Yoon et al., 2013*; [9] *Hansson et al., 2013*; [10] *Kariuki et al., 2013*; [11] *Eid et al., 2010*.
DOI: https://doi.org/10.7554/eLife.31579.004

seen in individuals of normal α-globin genotype (*Figure 5*). We found no evidence for an association between *Sl2* and any other clinical form of severe malaria (*Table 1*), or with *P. falciparum* parasite density (*Figure 6*).

## The *McC^b* allele is associated with increased susceptibility to cerebral malaria and death in the Kenyan case-control study

The unadjusted analysis showed a borderline significant association between *McC^b* and increased susceptibility to severe malaria overall (OR 1.17; 1.00–1.25; p=0.056, *Supplementary file 1D*), and

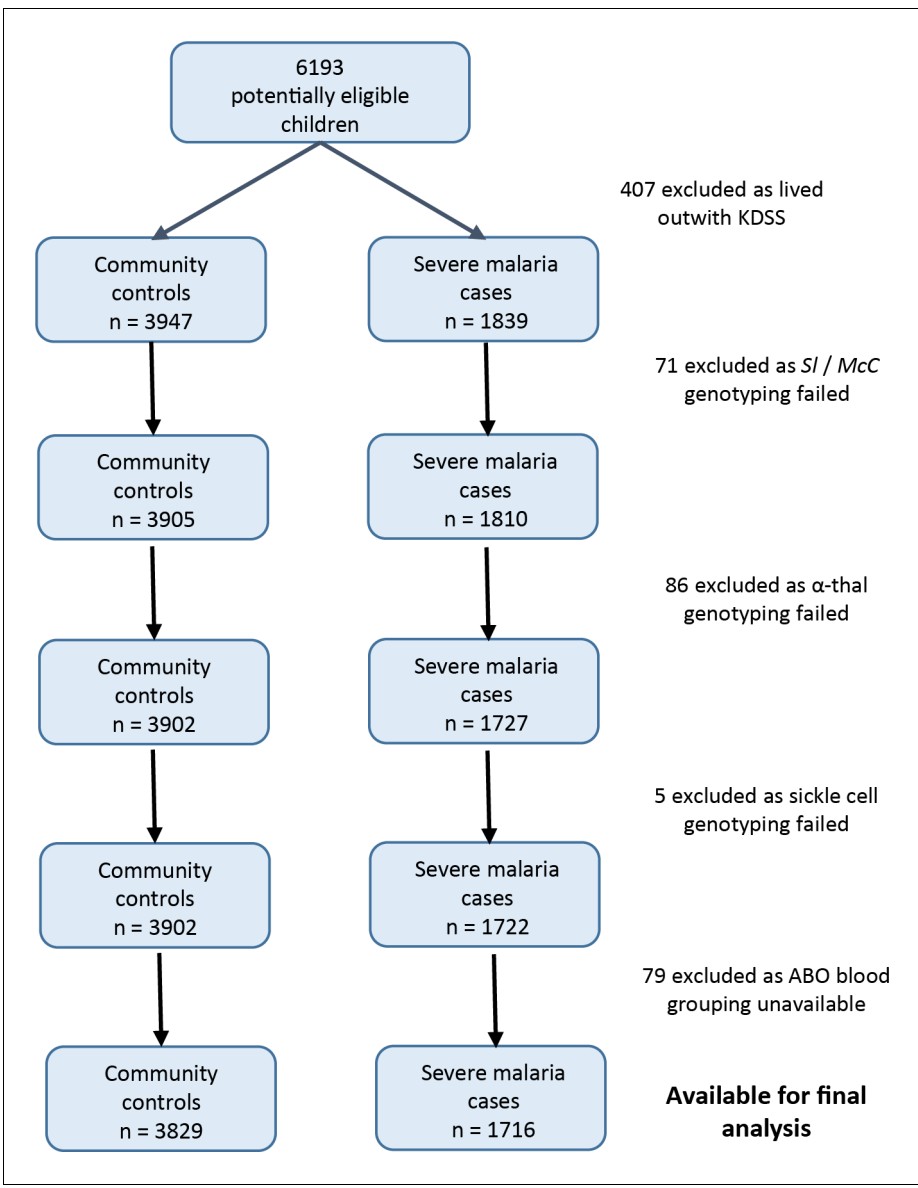

**Figure 3.** Patient inclusion flow chart for the Kenyan case-control study.
DOI: https://doi.org/10.7554/eLife.31579.005

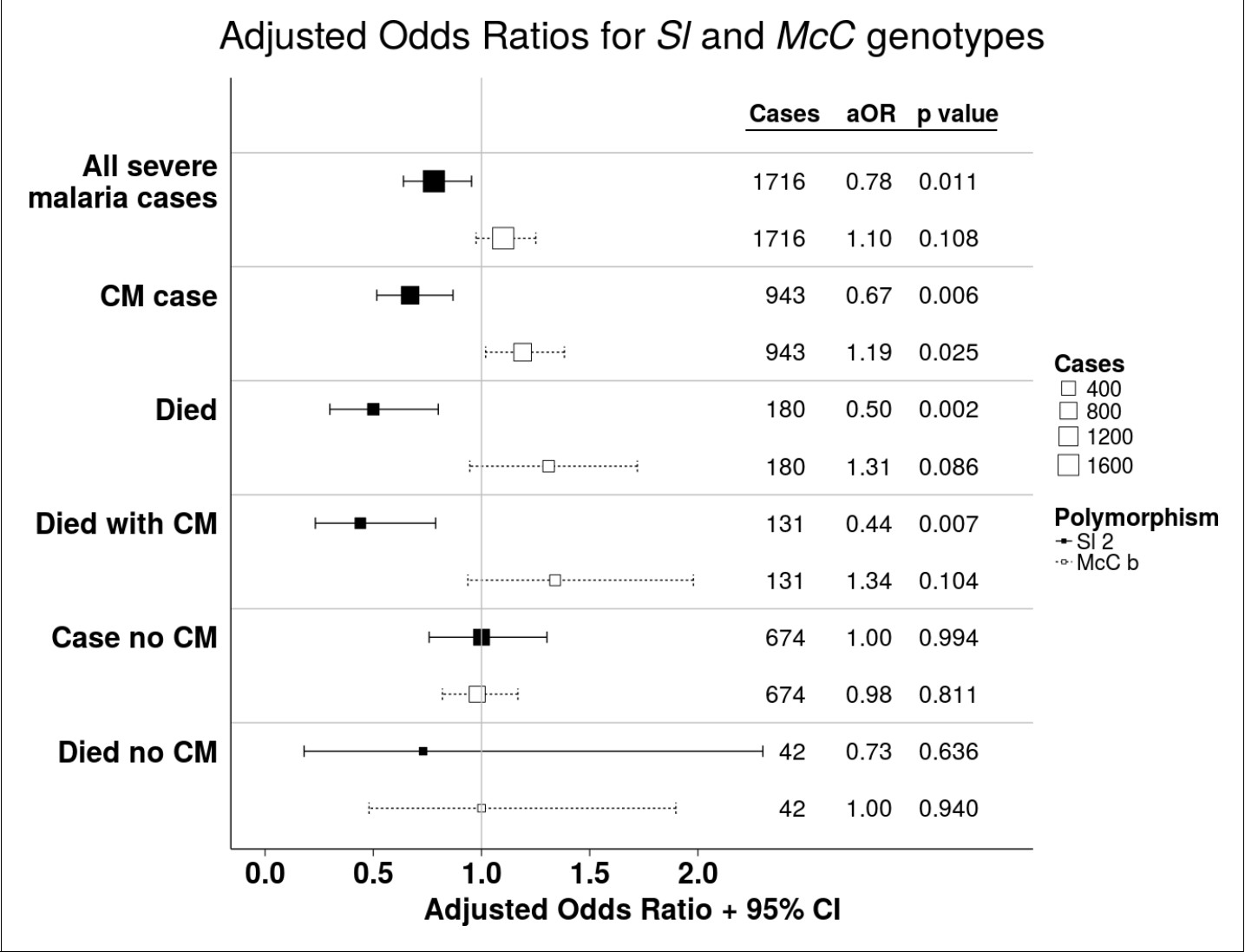

**Figure 4.** The *Sl2* and *McCb* alleles have opposing associations with cerebral malaria (CM) and death. Forest plot showing the associations between *Sl* and *McC* polymorphisms and severe malaria in Kilifi, Kenya. Filled boxes: adjusted Odds Ratios (aOR) for the *Sl2* genotype in the recessive form (i.e. *Sl2/Sl2* vs all other *Sl* genotypes). Open boxes: *McCb* in the additive form (i.e. change in odds ratio with each additional *McCb* allele). *Sl* and *McC* genotype were included together in a statistical model to examine their associations with malaria susceptibility. aORs displayed are adjusted for ethnicity, location of residence, sickle cell genotype, $\alpha^+$thalassaemia genotype and ABO blood group. An interaction term between *Sl* genotype and $\alpha^+$thalassaemia is included in the model. Model outputs following 2000 bootstrapped iterations are shown.

DOI: https://doi.org/10.7554/eLife.31579.006

significant associations with increased risk of cerebral malaria (OR 1.21; 1.05–1.39; p=0.008) and death (OR 1.34; 1.00–1.77; p=0.046, *Supplementary file 1D*). Similar associations were seen in the full adjusted analysis, although this only reached statistical significance for cerebral malaria (aOR 1.19; 1.10–1.38; p=0.025 (additive model), *Figure 4* and *Table 1*). We found no association between *McCb* and any other clinical form of severe malaria (*Table 1* and *Supplementary file 1D*) or with *P. falciparum* parasite density (*Figure 6*).

## Analysis of haplotypic effects and genotype combinations

We considered whether the observed results for *Sl* and *McC* could be consistent with the effect of a single haplotype spanning *Sl* and *McC*, or with the effect of a specific genotype combination. *Sl* and *McC* are 33 bp apart and are in linkage disequilibrium, with only three of four possible haplotypes

**Table 1.** Adjusted Odds Ratios (aOR) for severe malaria by *Sl2* (recessive) and *McC*[b] (additive) genotype in Kenya.

| Clinical outcome | *Sl2* aORs (95% CI)* | *P* value | *McC*[b] aORs (95% CI) | *P* value |
|---|---|---|---|---|
| All severe malaria[†] (n = 1716) | 0.78 (0.64–0.95) | 0.011 | 1.10 (0.97–1.25) | 0.108 |
| CM[§] (n = 943) | 0.67 (0.52–0.87)[‡] | 0.006 | 1.19 (1.02–1.38) | 0.025 |
| Severe without CM (n = 674) | 1.00 (0.76–1.30) | 0.994 | 0.98 (0.82–1.17) | 0.811 |
| Died (n = 180)[†] | 0.50 (0.30–0.80)[‡] | 0.002 | 1.31 (0.95–1.72) | 0.086 |
| Died with CM (n = 131) | 0.44 (0.23–0.78)[‡] | 0.007 | 1.34 (0.94–1.88) | 0.104 |
| Died without CM (n = 42) | 0.73 (0.18–2.30) | 0.636 | 1.00 (0.48–1.94) | 0.940 |
| SMA[#] (n = 483) | 0.76 (0.55–1.05) | 0.099 | 0.96 (0.78–1.17) | 0.688 |
| SMA without CM (n = 223) | 0.82 (0.51–1.26) | 0.366 | 0.91 (0.67–1.20) | 0.553 |
| Died with SMA[¶] (n = 56) | 0.65 (0.21–1.67) | 0.374 | 1.35 (0.77–2.20) | 0.229 |
| RD** (n = 522) | 0.81 (0.59–1.10) | 0.181 | 1.12 (0.92–1.35) | 0.225 |
| RD without CM (n = 192) | 1.06 (0.66–1.68) | 0.805 | 1.07 (0.80–1.43) | 0.615 |
| Died with RD[††] (n = 73) | 0.39 (0.14–0.88)[‡] | 0.027 | 1.01 (0.59–1.61) | 0.948 |

*Adjusted Odds Ratios (aOR) and 95% Confidence Intervals (CI) are presented for the *Sl2* genotype in the recessive form (i.e. *Sl2/Sl2* vs all other *Sl* genotypes) and *McC*[b] genotype in the additive form (i.e. change in aOR with each additional *McC*[b] allele). *Sl* and *McC* genotype were included together in a statistical model to examine their associations with malaria susceptibility. aORs displayed are adjusted for ethnicity, location of residence, sickle cell genotype, α[+]thalassaemia genotype and ABO blood group. An interaction term between *Sl* genotype and α[+]thalassaemia was included in the model. Model outputs following 2000 bootstrapped iterations are shown.

[†]99 children (7 of whom died) were severe malaria cases whose CM status was not recorded, hence these children are included in the numbers for 'All severe malaria' and 'Died' but not in 'with CM' or 'without CM' categories.

[‡]Models that showed significant evidence of interaction between *Sl2* and α[+]thalassaemia.

[§]CM, cerebral malaria (*P. falciparum* infection with a Blantyre coma score of < 3).

[#]SMA, severe malarial anaemia (*P. falciparum* infection with Hb < 5 g/dl).

[¶]34/56 cases who died with SMA also had CM.

**RD, respiratory distress (*P. falciparum* infection with abnormally deep breathing).

[††]56/73 cases who died with RD also had CM.

DOI: https://doi.org/10.7554/eLife.31579.009

observed in our data. We therefore reanalyzed the data under a haplotype model in which the per-individual count of each of the three observed haplotypes was included as a predictor along with the potential confounding factors, as well as under a genotypic model in which the count of each of the six possible *Sl/McC* genotype combinations was included as a predictor (Appendix 2). These analyses suggest an additive protective association with the *Sl2/McC*[a] haplotype (aOR = 0.85; 0.75–0.96; p=0.007), with broadly consistent results observed for analysis of genotype combinations (*Supplementary file 1E and 1F*). Thus, the opposing effects of *Sl2* and *McC*[b] observed above could plausibly result from the protective association of a single haplotype at the locus, although this is difficult to distinguish from the individuals SNPs acting independently and additively based on the statistical evidence alone.

### The *Sl2/Sl2* genotype was associated with protection against uncomplicated malaria in the Kenyan longitudinal cohort study

We next examined the association between *Sl2* and *McC*[b] alleles and uncomplicated malaria in a longitudinal prospective study of 208 Kenyan children. General characteristics of the cohort study population by *Sl* and *McC* genotypes are shown in *Supplementary file 1G*. After adjusting for variables known to influence malaria susceptibility, the *Sl2* allele was associated with a >50% reduction in the incidence of uncomplicated malaria (additive model) (*Table 2*; the number of episodes, incidence and unadjusted Incidence Rate Ratios for the diseases studied in the longitudinal cohort are shown in *Supplementary file 1H, I and J*). Once again, a significant interaction was seen with α[-+]thalassaemia, such that the protective association of *Sl2* was only demonstrated in children of normal α-globin genotype (*Table 3*). We found no significant association between the *McC*[b] allele and uncomplicated malaria (*Table 2*).

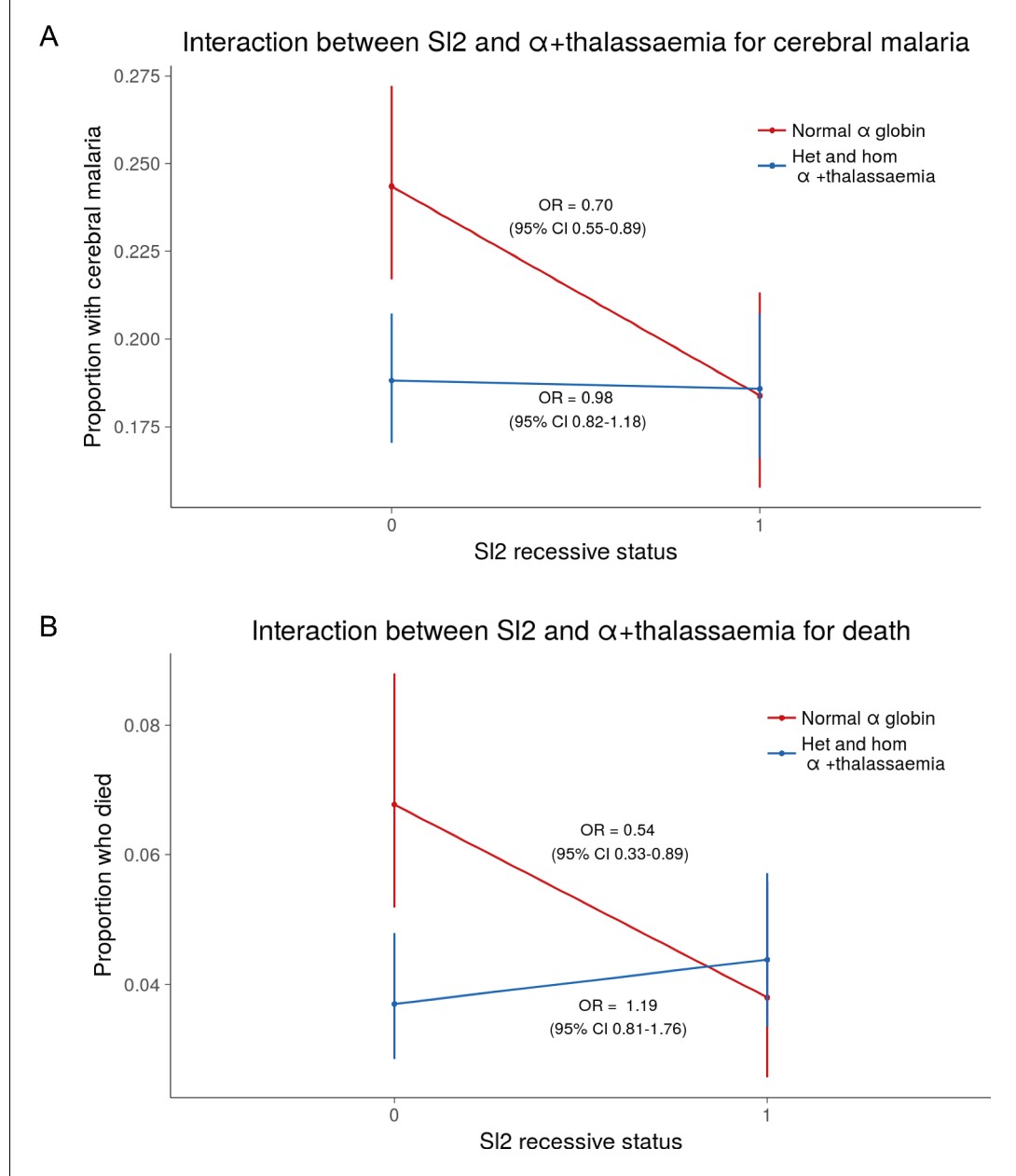

**Figure 5.** The protective association of *Sl2* with cerebral malaria and death is only evident in children with normal α-globin. Interaction plots showing the interaction between *Sl* (recessive) and α⁺thalassaemia for the proportion of children suffering (**A**) cerebral malaria and (**B**) death. For α⁺thalassaemia status, 0 = wild type α-globin; 1 = heterozygote or homozygote for α⁺thalassaemia. For *Sl* (recessive) status, 0 = *Sl1/Sl1* or *Sl1/Sl2* genotype; 1 = *Sl2/Sl2* genotype.

DOI: https://doi.org/10.7554/eLife.31579.007

## The *McC^b* allele was associated with protection from common non-malarial childhood diseases in the Kenyan longitudinal cohort study

The data shown above are incompatible with malaria being the selective pressure for *McC^b* in the Kenyan population, and suggest that other life-threatening childhood diseases may have been responsible for selection of *McC^b*. We therefore used the same longitudinal cohort study to investigate whether the *McC^b* and *Sl2* alleles influence the risk of other childhood diseases. *McC^b* was associated with borderline significant protection against several common infectious diseases including LRTIs, URTIs and gastroenteritis (*Table 2*). *Sl2* was associated with a borderline reduced

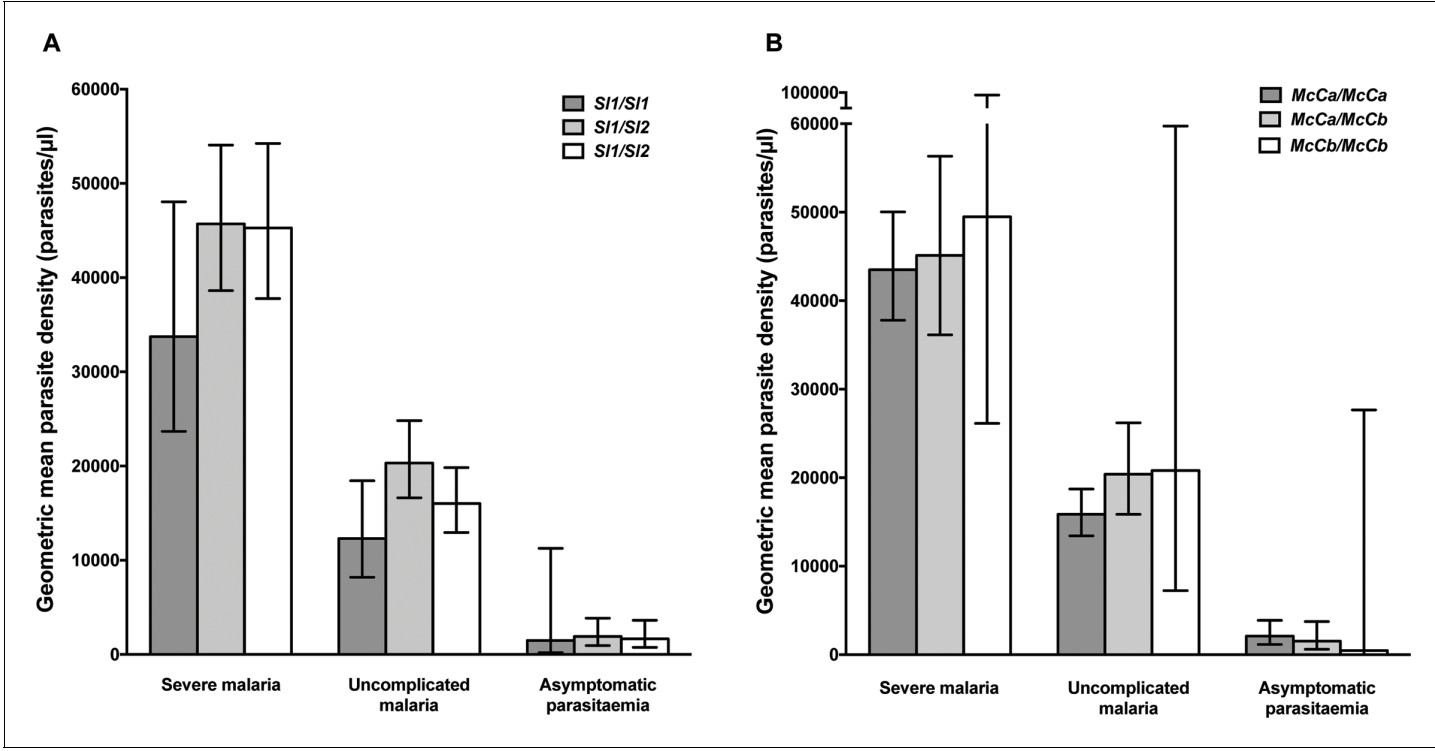

**Figure 6.** Parasite densities by *Sl* and *McC* genotypes. Geometric mean parasite densities in the Kenyan case-control study (severe malaria) and longitudinal disease cohort study (uncomplicated malaria and asymptomatic parasitaemia) by A) *Sl* genotypes and B) *McC* genotypes. The data on severe malaria includes 1695 children: (*Sl1/Sl1* (175), *Sl1/Sl2* (793), *Sl2/Sl2* (727) and *McC^a/McC^a* (1167), *McC^a/McC^b* (478) and *McC^b/McC^b* (50). The data on uncomplicated malaria includes 162 children: (*Sl1/Sl1* (16) , *Sl1/Sl2* (75), *Sl2/Sl2* (71) contributing 124, 488 and 461 episodes respectively and *McC^a/McC^a* (107), *McC^a/McC^b* (49) and *McC^b/McC^b* (6) contributing 699, 349 and 25 episodes, respectively. The data on asymptomatic parasitaemia includes 57 children: (*Sl1/Sl1* (5), *Sl1/Sl2* (26), *Sl2/Sl2* (26) contributing 6, 35 and 35 episodes, respectively, and *McC^a/McC^a* (34), *McC^a/McC^b* (20) and *McC^b/McC^b* (3) contributing 47, 25 and 4 episodes, respectively. Differences in parasite densities by genotype were tested by linear regression analysis with adjustment for HbAS, age as a continuous variable and ABO blood group in the severe malaria cases, HbAS and season (defined into 3-monthly blocks) in the uncomplicated malaria samples and HbAS and ABO blood group in the asymptomatic parasitaemia samples. Data were adjusted for within-person-clustering of events in the uncomplicated malaria and asymptomatic parasitaemia studies. Bars represent 95% confidence intervals.
DOI: https://doi.org/10.7554/eLife.31579.008

incidence of gastroenteritis (*Table 2*). The association of *McC^b* with gastroenteritis was predominantly seen in children of normal α-globin genotype, echoing the interaction seen with *Sl2* and malaria.

## The *Sl2* allele was associated with reduced ex vivo rosette frequency in *P. falciparum* clinical isolates from Mali

A previous in vitro study based on a culture-adapted *P. falciparum* parasite line suggested that RBC from *Sl2* genotype donors had a reduced ability to form rosettes, providing a possible mechanism for protection against severe malaria (*Rowe et al., 1997*). *P. falciparum* clinical isolates were not available from the Kenyan case-control study to investigate this potential mechanism in that population. However, the association of *Sl* and *McC* genotypes with ex vivo *P. falciparum* rosette frequency could be examined using 167 parasite isolates from a case-control study of children with clinical malaria in Mali (*Doumbo et al., 2009*). Analysis of this small case-control study suggested a protective association between the *Sl2/Sl2* genotype and cerebral malaria (aOR 0.35, 95% CI 0.12–0.89, p=0.024) and the *Sl2/Sl2-McC^a/McC^a* genotype combination was associated with protection against cerebral malaria (aOR 0.14, 95% CI 0.02–0.84, p=0.031, Appendix 1). As such, we considered samples from this population to be appropriate for testing rosetting as a potential mechanism of action. The median rosette frequency (percentage of iRBC that form rosettes) was significantly lower in *P. falciparum* isolates from malaria patients with one or more *Sl2* alleles than in isolates from *Sl1/Sl1*

**Table 2.** Adjusted Incidence Rate Ratios (aIRR) for uncomplicated malaria and non-malarial diseases in Kenya by *Sl* and *McC* genotype*.

| Clinical Outcomes | *Sl2* aIRRs[†] (95% CI) | P value | *McC*[b] aIRRs (95% CI) | P value |
|---|---|---|---|---|
| Uncomplicated malaria | **0.49 (0.34–0.72)[‡]** | **<0.001** [4] | 1.24 (0.90–1.70) | 0.184 [1] |
| All non-malaria clinical visits | 1.13 (0.96–1.32) | 0.140 [1] | **0.76 (0.61–0.96)[‡]** | **0.020** [4] |
| LRTI[§] | 1.09 (0.81–1.47) | 0.561 [1] | **0.39 (0.16–0.96)** | **0.040** [1] |
| URTI[#] | 1.21 (0.98–1.50) | 0.073 [1] | **0.79 (0.63–0.99)** | **0.047** [3] |
| Gastroenteritis | 0.66 (0.43–1.03) | 0.066 [2] | **0.55 (0.31–0.97)[‡]** | **0.038** [2] |
| Skin infection | 1.33 (0.79–2.26) | 0.285 [2] | 0.42 (0.16–1.13) | 0.086 [1] |
| Helminth infection | 1.98 (0.83–4.71) | 0.122 [2] | 0.68 (0.43–1.07) | 0.094 [4] |
| Malaria negative fever | 0.83 (0.58–1.18) | 0.293 [2] | 1.03 (0.80–1.33) | 0.828 [3] |

*Data were collected from 22 *Sl1/Sl1*, 94 *Sl1/Sl2* and 92 *Sl2/Sl2* individuals during 49.4, 213.8 and 188.8 cyfu (child-years of follow-up), respectively, and 137 *McC*[a]/*McC*[a], 63 *McC*[a]/*McC*[b] and 8 *McC*[b]/*McC*[b] individuals during 294.5, 143.2 and 14.3 cyfu, respectively. Both *Sl2* and *McC*[b] alleles were tested for their association with the disease outcomes of interest using Poisson regression in the [1]recessive, [2]dominant, [3]heterozygous and [4]additive models. The best fitting models as examined using the Akaike information criterion (AIC) were used in the final analysis that included adjustment for *McC* genotype (for *Sl* analyses), *Sl* genotype (for *McC* analyses) $\alpha^+$thalassaemia and sickle cell genotype, ABO blood group, season (divided into 3 monthly blocks), ethnicity, age as a continuous variable and within-person clustering of events.

[†]aIRRs: adjusted Incidence Rate Ratios.

[‡]Models that showed significant evidence of interaction between either *Sl2* or *McC*[b] and $\alpha^+$thalassaemia.

[§]LRTI: Lower Respiratory Tract Infection.

[#]URTI: Upper Respiratory Tract Infection.

DOI: https://doi.org/10.7554/eLife.31579.010

donors (*Figure 7*), whereas *McC* genotype had no significant associations with *P. falciparum* rosette frequency (*Figure 7*).

## Discussion

The data presented here provide epidemiological evidence supporting a role for CR1 in the pathogenesis of cerebral malaria. Two neighboring CR1 polymorphisms belonging to the Knops blood group system of antigens had opposing associations on risk of cerebral malaria. The *Sl2/Sl2* genotype was associated with protection against cerebral malaria and death, while the *McC*[b] allele was associated with increased susceptibility (*Figure 4* and *Table 1*). The *Sl2* allele was also associated with significant protection against uncomplicated malaria, whereas the *McC*[b] allele was associated with borderline protection against several common infections in Kenyan children (*Table 2*). The protective association of *Sl2* against cerebral malaria, death and uncomplicated malaria was influenced by $\alpha^+$thalassaemia, being most evident in children of normal $\alpha$-globin genotype.

**Table 3.** Incidence of uncomplicated malaria by *Sl* genotype and $\alpha$+thalassaemia status in the Kenyan longitudinal cohort study.

| | *Sl1/Sl1* | | *Sl1/Sl2* | | *Sl2/Sl2* | |
|---|---|---|---|---|---|---|
| | Number of episodes | Incidence | Number of episodes | Incidence | Number of episodes | Incidence |
| All samples | 124 | 2.51 | 493 | 2.31 | 461 | 2.44 |
| Normal $\alpha$ globin | 73 | 4.18 | 238 | 2.87 | 77 | 1.64 |
| Heterozygous $\alpha^+$thalassaemia | 32 | 1.58 | 209 | 1.92 | 302 | 2.88 |
| Homozygous $\alpha^+$thalassaemia | 19 | 1.63 | 46 | 2.09 | 82 | 2.20 |

Incidence = number of episodes per child-year of follow up (cyfu). Data were collected from 22 *Sl1/Sl1*, 94 *Sl1/Sl2* and 92 *Sl2/Sl2* individuals during 49.4, 213.8 and 188.8 child-years of follow-up, respectively.

DOI: https://doi.org/10.7554/eLife.31579.011

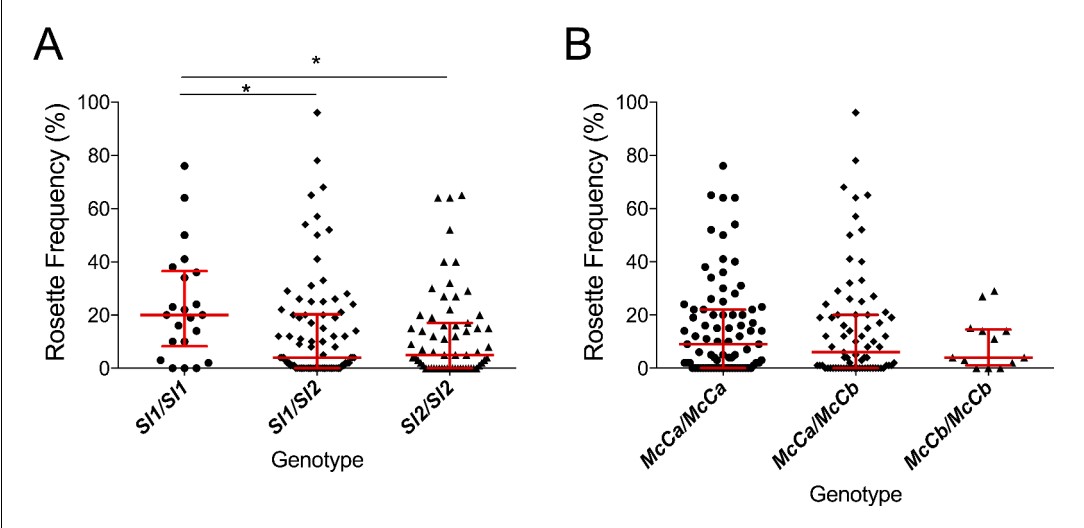

**Figure 7.** The *Sl2* allele is associated with reduced ex vivo rosette frequency of *P. falciparum* clinical isolates. Parasite isolates were collected from 167 malaria patients in Mali and matured in culture for 18–36 hr before assessment of rosette frequency (percentage of infected erythrocytes forming rosettes with two or more uninfected erythrocytes). Red bars show the median rosette frequency and interquartile range (IQR) for each genotype. (**A**) Rosetting by patient *Sl* genotype. *Sl1/Sl1* (n = 22, median 20.0, IQR 8.3–36.5), *Sl1/Sl2* (n = 82, median 4.0, IQR 0–20.3), *Sl2/Sl2* (n = 63, median 5.0, IQR 0–17.0); *p<0.05, Kruskal Wallis with Dunn's multiple comparison test; (**B**) Rosetting by *McC* genotype. *McC$^a$/McC$^a$* (n = 81, median 9.0, IQR 0–22.0), *McC$^a$/McC$^b$* (n = 73, median 6.0, IQR 0–20.0), *McC$^b$/McC$^b$* (n = 13, median 4.0, IQR 1–14.5); not significant, Kruskal Wallis with Dunn's multiple comparison test.

DOI: https://doi.org/10.7554/eLife.31579.012

The protective association between *Sl2* and cerebral malaria was first reported in a small case-control study from western Kenya (*Thathy et al., 2005*), but has remained controversial, especially as most prior studies have been underpowered. Hence, our study is the first adequately powered independent sample set that replicates the protective association between *Sl2* and cerebral malaria. Other studies found no consistent significant associations between *Sl* genotypes and severe malaria (*Zimmerman et al., 2003*; *Hansson et al., 2013*; *Jallow et al., 2009*; *Manjurano et al., 2012*; *Toure et al., 2012*; *Rockett et al., 2014*), including a recent multi-centre candidate gene study that included the sample set analysed here (*Rockett et al., 2014*). A weak association between *McC$^b$* and an increased odds ratio for cerebral malaria was shown in the multi-centre study (*Rockett et al., 2014*).

The complex interactions between *Sl2*, *McC$^b$* and α$^+$thalassaemia revealed by our study provide possible reasons for the previous inconsistent findings. Although *Sl2* was associated with protection against cerebral malaria in our study, *McC$^b$* and α$^+$thalassaemia both counteracted this effect. The protective association of *Sl2* was observed most clearly when both *McC$^b$* and α$^+$thalassaemia genotypes were included in the statistical model, something that has not been considered in previous studies. It is possible that some of the other discrepant genetic associations with severe malaria (*Rockett et al., 2014*) might result from interactions between multiple loci that vary across populations and may not be revealed by standard analyses. Biologically, it makes sense to account for *McC* genotype when investigating associations with *Sl2* and vice versa, as the two polymorphisms encode changes only 11 amino acids apart in the CR1 molecule (*Figure 1*). The possibility that the observed association might be due to a haplotype rather than independent effects of *Sl* and *Mc* cannot be discounted.

The interaction we describe here between *Sl2* and α$^+$thalassaemia is reminiscent of the epistatic interactions that have been observed between α$^+$thalassaemia and other malaria-protective polymorphisms including sickle cell trait (*HbAS*) (*Williams et al., 2005a*) and haptoglobin (*Atkinson et al., 2014*). It is possible, therefore, that α$^+$thalassaemia has a broad effect on multiple malaria-protective polymorphisms, influencing their restricted global frequencies (*Penman et al., 2009*), and contributing to the discrepant outcomes of previous association studies. Recent large

genetic association studies on malaria do not include data on $\alpha^+$thalassaemia, because the causal deletions are not typed on automated platforms (*Rockett et al., 2014*), instead requiring manual genotyping using labour-intensive PCR-based methods (*Chong et al., 2000*). Replication of the *Sl2/*$\alpha^+$thalassaemia interaction will be required, and we suggest that $\alpha^+$thalassaemia genotype should be included as an important confounding variable in future malaria epidemiological studies and that efforts should continue to discover the mechanism of protection afforded by $\alpha^+$thalassaemia, which remains controversial (*Carlson et al., 1994*; *Fowkes et al., 2008*; *Krause et al., 2012*; *Opi et al., 2014*; *Opi et al., 2016*).

We examined one possible biological mechanism by which the *Sl2* allele might influence cerebral malaria by studying *P. falciparum* rosetting, a parasite virulence factor associated with severe malaria in African children (*Doumbo et al., 2009*). Previous in vitro experiments showed that CR1 is a receptor for *P. falciparum* rosetting on uninfected RBCs, and that RBCs serologically typed as negative for the Sl1 antigen (likely to be from donors with *Sl1/Sl2* or *Sl2/Sl2* genotypes) (*Moulds et al., 2001*) show reduced binding to the parasite rosetting ligand PfEMP1 (Rowe et al., 1997) . In this study, we found a significantly lower median rosette frequency in *P. falciparum* parasite isolates from Malian patients with *Sl2* genotypes compared to *Sl1/Sl1* controls (*Figure 4*). Therefore, similar to HbC (*Fairhurst et al., 2005*), blood group O (*Rowe et al., 2007*) and RBC CR1 deficiency (*Cockburn et al., 2004*), it is possible that reduced rosetting and subsequent reduced microvascular obstruction (*Kaul et al., 1991*) may in part explain the protective association of *Sl2* against cerebral malaria. However, given the protective association of *Sl2* with uncomplicated malaria, and the possible associations of *Sl2* and *McC$^b$* with other common childhood infections, it seems likely that the Knops polymorphisms may be associated with broader effects, for example on the complement regulatory functions of CR1. Previously, we have shown that neither cofactor activity for the breakdown of C3b and C4b nor binding to C1q are influenced by the *Sl2* and *McC$^b$* mutations (*Tetteh-Quarcoo et al., 2012*). In addition, we can find no association between Knops genotype and CR1 clustering on erythrocytes (*Paccaud et al., 1988*; *Swann et al., 2017*). However, other potential effects such as altered immune complex binding and processing or activation of the complement lectin pathway via mannose-binding lectin (*Ghiran et al., 2000*) have not yet been investigated.

Our studies have several limitations: *McC$^b$* homozygotes are relatively infrequent in Kenya, which limited our power to detect associations with *McC$^b$* in the homozygous state. Our longitudinal cohort study generated several values of borderline statistical significance for the *McC$^b$* allele which are inconclusive. Studies with larger sample sizes will be needed to examine the specific associations of *McC$^b$* on assorted childhood diseases. Another limitation is that our functional (Mali) and epidemiological (Kenya) studies were conducted in different populations. The mechanisms of rosetting and associations with malaria severity are thought to be similar across sub-Saharan Africa (*Rowe et al., 2009*), suggesting that data collected in either location are likely to be comparable. Furthermore, examination of a small set of cerebral malaria cases and controls from Mali suggests a protective association between *Sl2/Sl2* genotype and cerebral malaria also occurs in this setting (Appendix 1). Ideally, future epidemiological and functional studies of specific polymorphisms on malaria should be conducted within a single population, although this remains logistically challenging.

In conclusion, we show that two high frequency CR1 polymorphisms have opposing associations with cerebral malaria and death in Kenyan children. While the *Sl2* allele may have reached high frequency in African populations by conferring a protective advantage against cerebral malaria, our data suggest that *McC$^b$* arose due to a survival advantage afforded against other non-malarial infections (*Noumsi et al., 2011*; *Fitness et al., 2004*). *Sl2* may in part protect against cerebral malaria by reducing rosetting, but additional effects seem likely. Further work is needed to examine both the epidemiological effects of the Knops polymorphisms on diverse childhood diseases, and the biological effects of the *Sl2* and *McC$^b$* polymorphisms on CR1 function. Future epidemiological studies should account for the effect of $\alpha^+$thalassaemia on the associations between *Sl2* and *McC$^b$* on malaria and other infectious diseases.

## Materials and methods

### Datasets studied

This study uses data from a Kenyan case-control study of severe malaria, with samples collected between 2001 and 2010, a Kenyan longitudinal cohort study, with samples collected between 1998 and 2001 and a Malian case-control study performed between July 2000 and December 2001. Historic datasets (i.e. >10 years old) are widely used in genetic epidemiological studies of malaria due to the logistical challenges of sample collection in malaria endemic countries and the changing epidemiological patterns of disease.

### The Kenyan study area

All epidemiological and clinical studies in Kenya were carried out in the area defined by the Kilifi Health and Demographic Surveillance System (KHDSS), with Kilifi County Hospital (KCH) serving as the primary point of care (*Scott et al., 2012*). Malaria transmission is seasonal in this region following the long and short rains. An Entomological Inoculation Rate (EIR) of up to 50 infective bites per person per year was measured in the late 1990s (*Mbogo et al., 2003*), but transmission has since declined (*O'Meara et al., 2008*).

### The Kenyan case-control study

Between January 2001 and January 2008, children aged <14 years who were admitted to KCH with severe malaria were recruited as cases, as described previously (*Rockett et al., 2014*), except that children who were resident outside the KHDSS were excluded (*Figure 3*). Severe malaria was defined as the presence of blood-film positive *P. falciparum* infection complicated by one or more of the following features: cerebral malaria (CM) (a Blantyre coma score (BCS) of <3) n = 943; severe malarial anaemia (SMA) (hemoglobin concentration of <5 g/dl) n = 483; respiratory distress (RD) (abnormally deep breathing) n = 522 or 'other severe malaria' (no CM, SMA or RD but other features including prostration (BCS 3 or 4), hypoglycemia and hyperparasitemia) n = 318. Controls (n = 3829) consisted of children 3–12 months of age who were born consecutively within the KHDSS study area between August 2006 and September 2010 and were recruited to an ongoing genetic cohort study (*Williams et al., 2009*). As such, controls were representative of the general population in terms of ethnicity and residence but not of age. The use of controls who are considerably younger than cases differs from the classical structure of a case-control study. However, this method (using cord blood or infant samples as controls) has been widely used in African genetic association studies (e.g. [*Band et al., 2013*; *Busby et al., 2016*; *Clarke et al., 2017*]) and is the most logistically feasible way of collecting sufficiently large numbers of control samples in many sub-Saharan African settings.

### Sample processing and quality control for the Kenyan case-control study

The *Sl* and *McC* polymorphisms were originally typed as part of a larger study by *Rockett et al., 2014*, which included case-control data from 12 global sites. In Kenya, 0.5 ml blood samples were collected into EDTA tubes and DNA extracted using Qiagen DNeasy blood kits (Qiagen, Crawley, UK). DNA was stored at −20°C and shipped frozen to Oxford. Sample processing is described in detail in the supplementary methods of *Rockett et al., 2014*. Briefly, samples underwent a whole-genome amplification step using Primer-Extension Pre-Amplification. Genotyping was performed using SEQUENOM iPLEX Gold with 384 samples processed per chip. In Rockett et al.'s study, samples were typed for 73 SNPs; 55 of these SNPs were chosen on the basis of a known association with severe malaria, 3 SNPs were used to confirm gender and the remaining 15 SNPs to aid quality control. Samples were excluded if they did not have clinical data for gender or if genotypic gender of the sample did not match clinical gender. Samples were included if they were successfully genotyped for more than 90% of 65 'analysis' SNPs. The Kenyan samples studied by Rockett et al. originally comprised 2741 cases of severe malaria and 4183 controls. After the quality control of both phenotypic and genotypic data described above, 2268 cases and 3949 controls were analysed by *Rockett et al., 2014*.

### Comparison between this study and *Rockett et al., 2014*.

The 2268 Kenyan cases and 3949 controls that were analyzed by *Rockett et al., 2014* were the starting point for our study. Children living outside the KHDSS were excluded, because this allowed us to use 'location' as a random effect in the final statistical model, which greatly improved model fit. Children with missing genotypes (*Sl, McC*, sickle cell, α$^+$thalassaemia or ABO blood group) were also excluded (*Figure 3*). After applying these exclusion criteria, 1716 severe malaria cases and 3829 community controls were available for analysis.

Hence, the number of severe malaria cases differs between our study and *Rockett et al., 2014* due to differing exclusion criteria. The inclusion of the severe malaria cases who lived outside the KHDSS into our statistical models did not alter the findings of our analysis (*Supplementary file 1K*). In both our study and *Rockett et al., 2014*, the control samples were identical and all came from within the KHDSS. Our study has 120 fewer controls than *Rockett et al., 2014* due to missing genotypes, because we only used controls for whom full *Sl, McC*, sickle cell genotype, α$^+$thalassaemia genotype and ABO blood group data were available.

Our analytical methods differed from *Rockett et al., 2014*, in that we included both *Sl* and *McC* in the same statistical model and adjusted for confounders, whereas Rockett et al. examined each SNP independently.

### The Kenyan longitudinal cohort study

This study has been described in detail previously (*Nyakeriga et al., 2004*). Briefly, this study was established with the aim of investigating the immuno-epidemiology of uncomplicated clinical malaria and other common childhood diseases in the northern part of the KHDSS study area, approximately 15 km from KCH (*Williams et al., 2005b*). The study was carried out between August 1998 and August 2001 involving children aged 0–10 years recruited either at the start of the study or at birth when born into study households during the study period. They were actively followed up on a once-weekly basis for both malaria and non-malaria related clinical events. In addition, on presentation with illnesses, cohort members were referred to a dedicated outpatient clinic for more detailed diagnostic tests. The cohort was monitored for the prevalence of asymptomatic *P. falciparum* infection through four cross-sectional surveys carried out in March, July and October 2000 and June 2001. Exclusion criteria included migration from the study area for more than 2 months, the withdrawal of consent and death. Uncomplicated clinical malaria was defined as fever (axillary temperature of > 37.5°C) in association with a *P. falciparum* positive slide at any density. The most common non-malaria-related clinical events reported during the study period included upper respiratory tract infections (URTIs), lower respiratory tract infections (LRTIs), gastroenteritis, helminth infections and skin infections, as defined in detail previously (*Williams et al., 2005b*). Malaria negative fever was defined as an axillary temperature of > 37.5°C in association with a slide negative for *P. falciparum*. This analysis includes 208 children aged < 10 years for whom full *Sl, McC*, sickle cell genotype, α$^-$$^+$thalassaemia genotype and ABO blood group data were available.

### The Malian case-control study

This study has been described in detail previously (*Lyke et al., 2003*). Briefly, between July 2000 and December 2001, children ranging from 1 month to 14 years of age were recruited into a case-control study in the Bandiagara region in East Central Mali, an area of intense and seasonal *P. falciparum* malaria infection. In order to address the specific question of whether the *Sl2/Sl2* genotype is associated with protection against cerebral malaria in Mali, only the subset of children suffering strictly defined cerebral malaria (a BCS of <3, with other obvious causes of coma excluded, n = 34) or uncomplicated malaria (n = 184, symptomatic children with *P. falciparum* parasitemia and an axillary temperature ≥37.5°C, in the absence of other clear cause of fever), and for whom *Sl* and *McC* genotyping was available were analyzed.

### Ex vivo rosetting

The rosette frequency (percentage of mature infected erythrocytes forming rosettes with two or more uninfected erythrocytes) of *P. falciparum* isolates from patients recruited into the Mali case-control study was determined by microscopy after short term culture (18–36 hr), as described in detail previously (*Doumbo et al., 2009*). Of the 209 isolates studied previously (*Doumbo et al.,*

*2009*), 167 were successfully genotyped for the *Sl* and *McC* alleles and are analysed here. The rosetting assays were performed before we genotyped the study participants, excluding observer bias. The rosette frequency of parasites from hosts with differing *Sl* and *McC* genotypes were compared by a Kruskal-Wallis test with Dunn's multiple comparisons (Prism v6.0, Graphpad Inc, San Diego, CA).

## Laboratory procedures

DNA was extracted either from fresh or frozen whole blood by proprietary methods using either the semi-automated ABI PRISM 6100 Nucleic acid prep station (Applied Biosystems, Foster City, CA) or using QIAamp DNA Blood Mini Kits (Qiagen, West Sussex, UK). SNPs giving rise to the *Sl* and *McC* alleles were genotyped using either the SEQUENOM iPLEX Gold multiplex system (Agena Biosciences, Hamburg, Germany) (Kenyan study) (*Rockett et al., 2014*) or by an established PCR-RFLP method as described previously (Malian study) (*Moulds et al., 2004*). Genotyping for sickle cell trait (HbAS) and the common African $\alpha^+$thalassaemia variant caused by a 3.7 kb deletion in the *HBA* gene were performed by PCR as described in detail elsewhere (*Chong et al., 2000*; *Waterfall and Cobb, 2001*).

## Statistical analysis

The effects of the *Sl* and *McC* alleles were examined in genotypic, dominant, recessive and additive models of inheritance, with the best fitting model selected based on Akaike information criterion (AIC). Analyses for the Kilifi case-control study were performed in R (R Foundation for Statistical Computing, Vienna, Austria) (*R Development Core Team, 2010*) using the 'ggplot2', 'lme4', and 'HardyWeinberg' packages (*Wickham, 2009*; *Bates et al., 2015*; *Graffelman and Camarena, 2008*), while analyses for the longitudinal study were performed in Stata v11.2 (StataCorp, Texas, USA). In both studies, a p value of < 0.05 was considered statistically significant. Graphs were generated using R or Prism v6.0 (Graphpad Inc, San Diego, CA).

For the Kenyan case-control study, *Sl* and *McC* genotype were included together in a statistical model to examine their associations with malaria susceptibility. Odds Ratios (ORs) and 95% Confidence Intervals (CI) were generated using mixed effect logistic regression analysis both with and without adjustment for ethnicity and location of residence as random effects, and sickle cell genotype, $\alpha^+$thalassaemia genotype, and ABO blood group (O or non-O) as fixed effects (variables which have been associated with malaria susceptibility in multiple previous studies in this population) (*Jallow et al., 2009*; *Rockett et al., 2014*; *Williams et al., 2005a*; *Atkinson et al., 2014*; *Rowe et al., 2007*; *Williams et al., 2005b*; *Fry et al., 2008*; *Malaria Genomic Epidemiology Network et al., 2015*). The ethnicity variable was compressed from 28 categories to four; Giriama (n = 2728), Chonyi (n = 1800), Kauma (n = 588) and other (n = 429). Binary parameterization of the $\alpha^-$$^+$thalassaemia variable was used, that is, comparing those children with no $\alpha^+$thalassaemia alleles against those with one or more $\alpha^+$thalassaemia alleles. This division was chosen in accordance with a previous report showing that both heterozygous and homozygous $\alpha^+$thalassaemia genotypes are associated with protection against severe malaria and death in the Kilifi area (*Williams et al., 2005c*). 2000 bootstrapped iterations were run to give 95% CIs and p values.

For the Kenyan longitudinal cohort study, Incidence Rate Ratios (IRRs) and 95% CIs were generated using a random effects Poisson regression model that took into account within-person clustering. Data were examined with and without adjustment for confounding by *McC* genotype (for *Sl* analyses), *Sl* genotype (for McC analyses) sickle cell genotype, $\alpha^+$thalassaemia genotype, ABO blood group, ethnic group, season (defined as 3 monthly blocks), and age in months as a continuous variable.

For the Malian case-control study, ORs and 95% CIs were computed using mixed effect logistic regression analysis with adjustment for location of residence as a random effect and age, ABO blood group (O or non-O) and ethnicity (Dogon or non-Dogon) as fixed effects. $\alpha^+$thalassaemia genotyping was not available for the Malian study and sickle cell trait is extremely uncommon in this population, therefore neither variable was included in the model. 2000 bootstrapped iterations were run to give adjusted ORs.

Corrections for multiple comparisons were not performed, instead all adjusted odds ratios, confidence intervals and p values have been clearly reported. This approach has been repeatedly

advocated, particularly when dealing with biological data (*Rothman, 1990*; *Perneger, 1998*; *Nakagawa, 2004*; *Fiedler et al., 2012*; *Rothman, 2014*). A detailed description of the Malian dataset is given in Appendix 1, and a detailed description of the statistical model fitting for the Kenyan studies is given in Appendix 2.

## Acknowledgements

We thank Johnstone Makale, Metrine Tendwa and Emily Nyatichi for laboratory support and all staff involved with data and sample collection at the Kilifi County Hospital, the KEMRI-Wellcome Trust Research Programme, Kilifi, Kenya and the Bandiagara Malaria Project Team, Bandiagara, Mali. We also thank the study participants and their parents for consenting to this study. This paper was published with permission from the Director of the Kenya Medical Research Institute (KEMRI).

## Additional information

### Funding

| Funder | Grant reference number | Author |
| --- | --- | --- |
| Wellcome | 203077 | D Herbert Opi |
| Wellcome | 084538 | D Herbert Opi |
| Wellcome | 101910/Z/13/Z | Olivia Swann |
| Medical Research Council | G19/9 | Dominic P Kwiatkowski |
| Wellcome | 091758 | Thomas N Williams |
| Wellcome | 202800 | Thomas N Williams |
| Wellcome | 084226 | J Alexandra Rowe |
| Wellcome | 067431 | J Alexandra Rowe |

The funders had no role in study design, data collection and interpretation, or the decision to submit the work for publication.

### Author contributions

D Herbert Opi, Olivia Swann, Data curation, Formal analysis, Investigation, Writing—original draft, Writing—review and editing; Alexander Macharia, Sophie Uyoga, Project administration, Writing—review and editing; Gavin Band, Data curation, Formal analysis, Validation, Writing—review and editing; Carolyne M Ndila, Data curation, Formal analysis, Writing—review and editing; Ewen M Harrison, Formal analysis, Writing—review and editing; Mahamadou A Thera, Abdoulaye K Kone, Dapa A Diallo, Ogobara K Doumbo, Kirsten E Lyke, Christopher V Plowe, Joann M Moulds, Conceptualization, Project administration, Writing—review and editing; Mohammed Shebbe, Neema Mturi, Norbert Peshu, Kathryn Maitland, Resources, Writing—review and editing; Ahmed Raza, Investigation, Writing—review and editing; Dominic P Kwiatkowski, Conceptualization, Resources, Data curation, Formal analysis, Funding acquisition, Validation, Investigation, Methodology, Writing—original draft, Project administration, Writing—review and editing; Kirk A Rockett, Conceptualization, Resources, Data curation, Formal analysis, Funding acquisition, Validation, Investigation, Methodology, Project administration, Writing—review and editing; Thomas N Williams, Conceptualization, Resources, Data curation, Supervision, Funding acquisition, Investigation, Methodology, Writing—original draft, Project administration, Writing—review and editing; J Alexandra Rowe, Resources, Supervision, Investigation, Writing—original draft, Writing—review and editing

### Author ORCIDs

D Herbert Opi http://orcid.org/0000-0002-4589-4365
Olivia Swann http://orcid.org/0000-0001-7386-2849
Mahamadou A Thera http://orcid.org/0000-0002-2679-035X
Kirk A Rockett https://orcid.org/0000-0002-6369-9299

Thomas N Williams [ID] http://orcid.org/0000-0003-4456-2382
J Alexandra Rowe [ID] http://orcid.org/0000-0002-7702-1892

### Ethics

Human subjects: This work involved analysing blood samples from patients with malaria and from healthy controls. Written informed consent was obtained from the parents or legal guardians of all participants. The studies received ethical approval from the Kenya Medical Research Institute National Ethical Review Committee (case control study: SCC1192; cohort study: SCC3149), the University of Bamako/Mali Faculty of Medicine, Pharmacy and Dentistry Institutional Review Board (which, at the time the study was accepted did not give out approval numbers) and the University of Maryland (approval number #0899139).

### Decision letter and Author response

Decision letter https://doi.org/10.7554/eLife.31579.024
Author response https://doi.org/10.7554/eLife.31579.025

## Additional files

### Supplementary files

• Supplementary file 1. Additional tables. (A) General characteristics for the Kenyan case-control study. (B) Characteristics for the Kenyan case-control study by *Sl* and *McC* genotype. (C) Hardy Weinberg equilibrium calculations for controls in the Kenyan case-control study. (D) Unadjusted odds ratios for clinical outcomes for the Kenyan case-control study. (E) *Sl* and *McC* combined genotypes and adjusted odds ratios for cerebral malaria in the Kenyan case-control study. (F) *Sl* and *McC* combined genotypes and adjusted odds ratio for death in the Kenyan case-control study. (G) General characteristics of the Kenyan longitudinal cohort study population by *Sl* and *McC* genotypes. (H) Incidence of common childhood diseases by *Sl* genotypes in the Kenyan longitudinal cohort study. (I) Incidence of common childhood diseases by *McC* genotypes in the Kenyan longitudinal cohort study. (J) Unadjusted Incidence Rate Ratios (IRR) for uncomplicated malaria and non-malarial diseases in the Kenyan longitudinal cohort study by *Sl* and *McC* genotype. (K) Reanalysis of Kenyan case-control study including children who lived outside of the KHDSS study area. (L) Adjusted Odds Ratios for different genetic models for the *Sl* polymorphism in the Kenyan case-control study. (M) Adjusted Odds Ratios for different genetic models for the *McC* polymorphism in the Kenyan case-control study. (N) Investigation of the sickle trait/$\alpha^+$thalassaemia negative epistatic interaction and the *Sl2*/$\alpha^+$thalassaemia interaction by clinical outcome in the Kenyan case-control study. (O) Reanalysis of the Kenyan case-control study excluding all children with one or more sickle cell alleles. (P) Raw data for the combined sickle trait, $\alpha^+$thalassaemia and *Sl* genotype by clinical outcome in the Kenyan case-control study. (Q) Correlations between the sickle cell, $\alpha$ + thalassaemia, *Sl2* and *McC$^b$* variants in the Kenyan case-control study. (R) Adjusted incidence Rate Ratios (IRRs) for *Sl* disease associations in the longitudinal cohort study by genetic models of inheritance (S) Adjusted incidence rate ratios for *McC* disease associations in the longitudinal cohort study by genetic models of inheritance

DOI: https://doi.org/10.7554/eLife.31579.013

• Reporting standard 1

DOI: https://doi.org/10.7554/eLife.31579.014

• Transparent reporting form

DOI: https://doi.org/10.7554/eLife.31579.015

### Major datasets

The following previously published dataset was used:

| Author(s) | Year | Dataset title | Dataset URL | Database, license, and accessibility information |
|---|---|---|---|---|
| MalariaGEN Consortium | 2014 | MalariaGEN Consortial Project 1 | https://www.malariagen.net/projects/consortial- | Application for access to data: https://www. |

project-1                                   malariagen.net/data/
                                            terms-use/human-
                                            gwas-data

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

## Appendix 1

DOI: https://doi.org/10.7554/eLife.31579.016

# Validation of the Mali case-control study as a source of samples to examine the effect of Knops genotype on *P. falciparum* rosetting

## The *Sl2/Sl2* genotype is associated with protection against cerebral malaria in Mali

To determine if the Mali case-control study was a suitable source of samples to examine the effect of Knops genotype on *P. falciparum* rosetting (*Doumbo et al., 2009*), we examined whether there was any evidence to suggest that the association of the *Sl2/Sl2* genotype with cerebral malaria also occurred in Mali. To do this, we examined the cerebral malaria cases (n = 34) and uncomplicated malaria controls (n = 184) from a case-control study (*Appendix 1—figure 1*) (*Lyke et al., 2003*). General characteristics of the cases and controls are shown in *Appendix 1—table 1* and general characteristics by *Sl* and *McC* genotype are shown in *Appendix 1—table 2* below.

**Patient inclusion flow chart
for Mali case-control study**

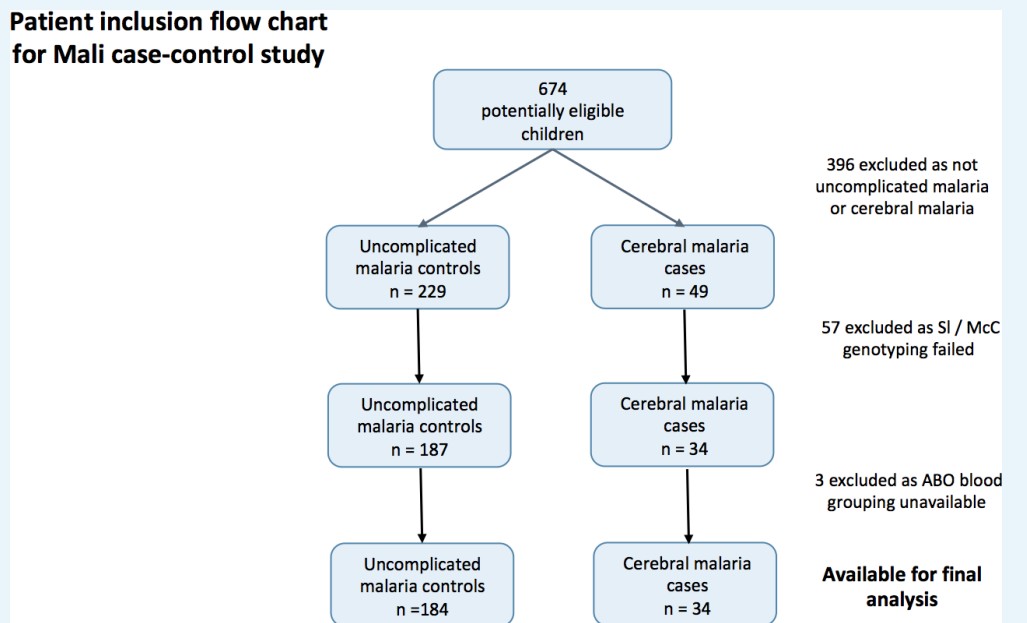

**Appendix 1—figure 1.** Patient inclusion flow-chart for the Mali case-control study.

DOI: https://doi.org/10.7554/eLife.31579.017

A mixed effect logistic regression analysis with adjustment for location of residence as a random effect and age, ABO blood group (O or non-O) and ethnicity (Dogon or non-Dogon) as fixed effects showed a protective association of the *Sl2/Sl2* genotype (recessive model) against cerebral malaria (aOR 0.35, 95% CI 0.12–0.89, p=0.024). *McC*[b] (additive model) did not show a statistically significant association with increased odds of cerebral malaria (aOR 1.53, 95% CI 0.77–3.20, p=0.212). *Sl2/Sl2-McC*[a]/*McC*[a] was the only combined *Sl/McC* genotype to be significantly associated with protection (aOR 0.14, 95% CI 0.02–0.84, p=0.031). $\alpha^+$thalassaemia genotype data were not available for the Mali samples to test for interaction. Therefore, given that the data do suggest a protective association between *Sl2/Sl2* genotype and cerebral malaria in Mali, we considered samples from this population to be appropriate for testing rosetting as a potential mechanism of action.

**Appendix 1—table 1.** General characteristics for cases and controls in the Mali case- control study.

| | Controls (Uncomplicated malaria) | Cases (Cerebral malaria) | P value |
|---|---|---|---|
| Mali | n = 184 | n = 34 | |
| Gender<br>Males<br>Females | 90 (49 %)<br>94 (51 %) | 17 (50 %)<br>17 (50 %) | 0.920 |
| Ethnicity<br>Dogon<br>Non-Dogon | 161 (87.5 %)<br>23 (12.5 %) | 30 (88 %)<br>4 (12 %) | 1 |
| Age in months*<br>Median (IQR) | 36.5 (19–56) | 28 (16–41) | 0.026 |

Comparisons performed using Pearson's $\chi^2$ test except *Kruskal Wallis test

DOI: https://doi.org/10.7554/eLife.31579.018

## Appendix 2

DOI: https://doi.org/10.7554/eLife.31579.020

# Detailed statistical methods

## Statistical model fitting and bootstrapping for the Kenyan case-control study

Analyses for the Kilifi case-control study were performed in R (R Foundation for Statistical Computing, Vienna, Austria) (*R Development Core Team, 2010*) using the 'ggplot2', 'lme4', and 'HardyWeinberg' packages (*Wickham, 2009*; *Bates et al., 2015*; *Graffelman and Camarena, 2008*). The dataset was restricted to children who were resident in the Kilifi Health and Demographic Surveillance System (KHDSS) (*Scott et al., 2012*) and had full genotyping data for *Sl*, *McC* $\alpha^+$thalassaemia, sickle cell and ABO blood group. This resulted in 1716 cases and 3829 controls (*Figure 3*). The ethnicity variable was compressed from 28 categories to four; Giriama (n = 2728), Chonyi (n = 1800), Kauma (n = 588) and other (n = 429). Binary parameterization of the $\alpha^+$thalassaemia variable was used, that is, comparing those children with no $\alpha^+$thalassaemia alleles against those with one or more $\alpha^+$thalassaemia alleles. This division was chosen in accordance with a previous report showing that both heterozygous and homozygous $\alpha^+$thalassaemia genotypes are associated with protection against severe malaria and death in the Kilifi area (*Williams et al., 2005c*).

A simple unadjusted logistic regression analysis containing only the *Sl* and *McC* genotypes suggested potential associations with severe malaria (*Supplementary file 1D*). We attempted to refine this signal by fitting a more complete model to the data using mixed effect logistic regression analysis. The full adjusted analysis was constructed as follows:

1. Variables associated with malaria susceptibility in multiple previous studies in this population (*Jallow et al., 2009*; *Rockett et al., 2014*; *Williams et al., 2005a*; *Atkinson et al., 2014*; *Rowe et al., 2007*; *Williams et al., 2005b*; *Fry et al., 2008*; *Malaria Genomic Epidemiology Network et al., 2015*) were included to give a 'base model'. These variables were sickle cell genotype (as a binary variable, sickle trait vs no sickle trait), $\alpha^+$thalassaemia genotype (as a binary variable, one or more $\alpha^+$thalassaemia alleles vs no $\alpha^+$thalassaemia alleles) and ABO blood group (as a binary variable, group O vs non-group O).
2. Both *Sl* and *McC* genotype were added in the simplest form (additive).
3. All possible models for *Sl* were then examined (genotypic, dominant, recessive, heterozygous and additive models of inheritance, see *Supplementary file 1L*).

Model selection was performed using a criterion-based approach by minimizing the Akaike information criterion (AIC) and discrimination (i.e. how well a model separates individuals with and without the outcome of interest) was determined using the c-statistic (area under the receiver operator curve). The recessive model had the best overall fit for *Sl* across clinical outcomes.

1. The process of examining all genetic models of inheritance was then repeated for *McC* (*Supplementary file 1M*), with *Sl* as recessive included in each model. The additive model had the best overall fit for *McC* across clinical outcomes.
2. All first order interactions were explored. The interaction with the greatest effect on the AIC was included in the final model. This resulted in an interaction term between *Sl2* and $\alpha^-{}^+$thalassaemia being incorporated into the model.
3. Finally, ethnicity and location of residence were incorporated as random effects in order to accommodate population structures which did not require quantification for this study.

The final model incorporated ethnicity and location of residence as random effects, and as fixed effects had *Sl2* in the recessive form (i.e. binary variable, *Sl2/Sl2* vs *Sl1/Sl1* or *Sl1/Sl2*); presence of at least one $\alpha^+$thalassaemia allele vs no $\alpha^+$thalassaemia alleles (binary variable);an interaction term between the *Sl* and $\alpha^+$thalassaemia variables; *McC*[b] in the additive form (i.e.

**Appendix 1—table 2.** Table of Characteristics for the Malian dataset by SI and McC genotype.

| | | SI1/SI1 | % | SI1/SI2 | % | SI2/SI2 | % | P value | McC$^a$/McC$^a$ | % | McC$^a$/McC$^b$ | % | McC$^b$/McC$^b$ | % | P value |
|---|---|---|---|---|---|---|---|---|---|---|---|---|---|---|---|
| Clinical status | Uncomplicated malaria | 19 | 10.3 | 78 | 42.4 | 87 | 47.3 | | 103 | 56.0 | 61 | 33.2 | 20 | 10.9 | |
| | CM | 5 | 14.7 | 19 | 55.9 | 10 | 29.4 | 0.135[#] | 16 | 47.1 | 15 | 44.1 | 3 | 8.8 | 0.478[#] |
| Gender | M | 11 | 10.3 | 48 | 44.9 | 48 | 44.9 | | 57 | 53.3 | 41 | 38.3 | 9 | 8.4 | |
| | F | 13 | 11.7 | 49 | 44.1 | 49 | 44.1 | 0.945 | 62 | 55.9 | 35 | 31.5 | 14 | 12.6 | 0.430 |
| Ethnicity | Dogon | 21 | 11.0 | 88 | 46.1 | 82 | 42.9 | | 108 | 56.5 | 66 | 34.6 | 17 | 8.9 | |
| | Non-Dogon | 3 | 11.1 | 9 | 33.3 | 15 | 55.6 | 0.448[#] | 11 | 40.7 | 10 | 37.0 | 6 | 22.2 | 0.078 |
| Blood group | O | 13 | 10.0 | 63 | 48.5 | 54 | 41.5 | | 68 | 52.3 | 49 | 37.7 | 13 | 10.0 | |
| | Non-O | 11 | 12.5 | 34 | 38.6 | 43 | 48.9 | 0.355 | 51 | 58.0 | 27 | 30.7 | 10 | 11.4 | 0.566 |
| Median age | Months | 32 | IQR 20–57.5 | 31 | IQR 18–52 | 30 | IQR 17–55 | 0.927 | 31 | IQR 17–52.5 | 51 | IQR 18–54.5 | 51 | IQR 29–64.5 | 0.229 |

CM, cerebral malaria; IQR, interquartile range. Significance testing uses uncomplicated malaria as the reference group. Age: Kruskal Wallis test. [#]: Fisher's exact test. All other analyses: Pearson's $\chi^2$.

DOI: https://doi.org/10.7554/eLife.31579.019

impact of each additional $McC^b$ allele); presence/absence of sickle cell trait as a binary variable; O/non-O blood group as a binary variable.

The code used for the final model was: fit = glmer (outcome ~ (1|ethnicity) + (1|location) +Sl_recessive*thalassaemia_allele + Mc + sickle_trait + non_O, data = data, family = binomial)

Bootstrapping was performed using the 'bootMer' function in package 'lme4' in R. 2000 iterations were run of each model to calculate 95% confidence intervals and p values. If models did not converge over these 2000 iterations they were inspected for singularities (i.e. a level of one of the variables having a value of 0, for example 0 cases living in Gede). If no singularities were identified, the bootstrapping was rerun using the optimiser 'bobyqa' with $10^5$ evaluations.

Corrections for multiple comparisons were not performed in this study, instead all adjusted odds ratios, confidence intervals and p values have been clearly reported. This approach has been repeatedly advocated, particularly when dealing with biological data (**Rothman, 1990**; **Perneger, 1998**; **Nakagawa, 2004**; **Fiedler et al., 2012**; **Rothman, 2014**). The stringency of multiple comparisons increases the risk of type II error, potentially discarding important findings. No single study can be considered conclusive and novel results will always require replication.

## Exploration of alternative haplotype and combined genotype models

As one of the four possible $Sl/Mc$ haplotype combinations was not seen ($Sl1/McC^b$), the $Sl2$ and $McC^b$ alleles are likely to be in complete linkage disequilibrium in this population sample (i.e. D'=1, no recombination between these two markers). This situation makes it difficult to distinguish statistically between a model where $Sl$ and $McC$ act independently and additively or a haplotype model. We considered the possibility that a haplotype model could provide an alternative explanation for our findings, with a separate true protective mutation being positively tagged by the $Sl2$ allele and negatively tagged by the $McC^b$ allele. Specifically, for each sample we computed the count of each of the three possible $Sl/Mc$ haplotypes (assuming only three haplotypes are segregating as above). We then re-fit the logistic regression model for cerebral malaria using haplotype counts as predictors, in addition to potential confounders included in the full adjusted analysis described above. This model estimates a non-zero protective effect of the $Sl2/McC^a$ haplotype (additive OR = 0.85; 0.75–0.96; p=0.007), but did not fit as well as the full adjusted analysis described above (AIC = 4268.5, versus 4266.8 for the full analysis).

As both $Sl$ and $Mc$ have sufficient structural effects to alter Knops blood group phenotype, it would appear reasonable to examine their function further before looking for other nearby mutations. In addition, no other strong effects near CR1 have been identified by GWAS studies that could explain the association. However, a haplotype model cannot be excluded as a possibility on the basis of our current data.

## Exploration of the negative epistasis between sickle trait and $\alpha^+$thalassaemia

Previous studies have described a negative epistatic interaction between sickle trait and $\alpha^+$thalassaemia, reporting that $\alpha^+$thalassaemia homozygotes who also carry the sickle trait are not protected from severe malaria (**Williams et al., 2005a**). We wanted to ensure that an unrecognised relationship between sickle cell trait and $Sl$ genotype did not account for the interaction between $\alpha^+$thalassaemia and $Sl$ that we report in the current study.

Analysis of our current dataset confirmed the existence of negative epistasis between sickle trait and $\alpha^+$thalassaemia in this population (**Supplementary file 1N**). However, of interest, this negative epistatic interaction was only seen in the severe malaria cases without cerebral malaria, whereas the $\alpha^+$thalassaemia/$Sl$ interaction was specific to cerebral malaria cases (**Supplementary file 1N**). Therefore, the two interactions appear to be mutually exclusive.

The final adjusted analysis was also re-run on a restricted dataset which excluded the 664 children with sickle cell trait or sickle cell disease. The results of this analysis remained

unchanged and the $\alpha^+$thalassaemia/*Sl* interaction persisted without any influence of sickle trait (**Supplementary file 1O**).

Sickle cell trait did not show a statistical interaction with either *Sl* or *McC* genotype. The sickle cell mutation is far less common in the KHDSS population than the $\alpha^+$thalassaemia mutation (~12% of children have one or more sickle cell alleles, compared to ~ 64% with one or more $\alpha^+$thalassaemia alleles). As such, even in a study as large as this one, the power to detect statistically significant interactions between all three of sickle, $\alpha^+$thalassaemia and *Sl* genotypes is greatly reduced. However, we found no evidence of a three way interaction between these alleles. The raw data for the combined genotypes compromising sickle trait, $\alpha^+$thalassaemia and *Sl* for each clinical outcome is presented in **Supplementary file 1P**. Correlations between sickle cell, $\alpha^+$thalassaemia, *Sl* and *McC* are presented in **Supplementary file 1Q**.

## Statistical model fitting for the longitudinal cohort study

Associations between *Sl* and *McC* and mild malaria and other non-malarial related diseases in the longitudinal cohort study were tested in Stata v11.2 (StataCorp, Texas, USA) using a random effects Poisson regression analysis that accounted for within-person clustering of events. This analysis was restricted to children under 10 years old living in the Ngerenya area in the northern part of the KHDSS study area. The analysis was carried out on the 208 children from the cohort with full genotype, ethnic group, season and age data. The model selection process first involved univariate analyses testing for disease associations for *Sl* and *McC* independently without potential confounders in genotypic, dominant, recessive, heterozygous and additive models of inheritance. Models were compared using the Akaike information criterion (AIC) for fitness, with the model displaying the minimum AIC values for each respective genotype and outcome of interest chosen as the best fitting model. The unadjusted Incident Rate Ratios and best fitting models are shown in **Supplementary file 1J**. For each disease outcome, the association with *Sl* genotype was then adjusted for confounding by *McC* (best genetic model chosen from the univariate analysis) and for explanatory variables previously associated with outcomes of interest: sickle cell genotype, $\alpha^+$thalassaemia genotype, ABO blood group genotype, ethnic group (Giriama, Chonyi and others), season (defined as 3 monthly blocks) and age in months as a continuous variable. AIC values were compared to identify the best fitting genetic model (**Supplementary file 1R**). The same process was carried out for the association of *McC* genotype with each disease outcome, with adjustment for *Sl* genotype and the other explanatory variables (**Supplementary file 1S**). For consistency of reporting here, the same explanatory variables are included in the statistical models for all disease outcomes in the data presented. Optimized model-fitting for each outcome by removing explanatory variables that did not improve model fit, did not make any material difference to the results shown here.

Finally, we tested for interactions between either *Sl* and *McC* and $\alpha^+$thalassaemia (represented as normal, heterozygous and homozygous genotypes) using the likelihood ratio test with a p value of <0.05 indicating statistically significant evidence for interaction, and the appropriate interaction term included in the final model.

