## [Decision Letter]

Thank you for submitting your article "Two complement receptor one alleles have opposing associations with cerebral malaria and interact with α^+^thalassaemia" for consideration by *eLife*. Your article has been favorably evaluated by Prabhat Jha (Senior Editor) and three reviewers, one of whom, Madhukar Pai (Reviewer #1), is a member of our Board of Reviewing Editors.

The reviewers have discussed the reviews with one another and the Reviewing Editor has drafted this letter to draw your attention to significant concerns that you would need to address before this work could be considered for publication.

Summary:

This is an interesting but complex study with different study designs, looking at interactions between 3 mutations and malaria. The study raises hypotheses that will need to be confirmed in future studies.

The study evaluated the relationship between 2 SNPs (rs17047661 and rs17047660) of *CR1* and several malaria phenotype (cerebral malaria, death, uncomplicated malaria other severe forms of malaria) as well as non-malarial disease. The authors found that 1) the *Sl2/Sl2* genotype is associated with protection against cerebral malaria and death, p=0.006 and p= 0.002 respectively, 2) the *McC^b^* allele is associated with increased susceptibility to cerebral malaria and death, p=0.008 and p=0.046 respectively, 3) the *Sl2/Sl2* genotype was associated with protection against uncomplicated malaria p<0.001, and 4) the *McC^b^* allele was associated with protection from non-malarial disease p=0.02. Finally, the authors demonstrated that the allele *Sl2* was associated with reduced ex vivo rosette frequency, whereas *McC* had no significant effect on *P. falciparum* rosette frequency. The paper is interesting and suggests the role of *CR1* polymorphisms in susceptibility/resistance to malaria and non-malarial disease. However, the reviewers have major concerns on several points.

Essential revisions:

1) Given the dramatic reductions in severe malaria and deaths in the past decade, it is a bit strange that the studies were done with data collected during 2001-08 (case control study), and 1998-2001 (cohort study). So, a decade later, the authors have gone back to analyze stored samples. The implications of this are not clear and need to be clearly addressed. Are there time trends that can confound the associations? What about use of old samples and sample quality? Were the hypotheses pre-stated or generated post-hoc? (presumably the latter).

2) As regards the case-control study, I would need more convincing that cases and controls were derived from the same study base. The difference in age distribution will likely not matter for the genetic mutations, but what about implications for disease severity?

3) The authors indicated that the study included 5545 children from Kenya consisting in 1716 severe malaria cases and 3829 community controls.a) The authors performed the analysis on a subset of the samples used previously by Rockett et al. (2014). The authors should clarify why they did not use the total sample set of 2268 severe malaria cases included in Rockett et al. and how the 1716 samples were selected.

b) As indicated by the authors, 407 individuals are excluded as lived out with KHDSS (Figure 3). Why? The authors should clarify this point. DNA not available? It would be surprising because all CR1 genotype (rs17047661 and rs17047660), sickle cell genotype, a^+^thalassaemia genotype and ABO blood group were available in the previous study of Rockett for the Kenyan population.

c) It is not clear for me whether new genotyping has been performed for this study or if they re-used the genotypes obtained in the Rockett's study.

d) Why did the authors use controls recruited to an ongoing genetic cohort that were representative of the general population (Williams et al., 2009) as indicated in the subsection “The Kilifi case-control study” rather than used the controls of the previous study (Rockett et al.) that were also representative of the general populations? Is there a scientific reason? Are the Rockett controls not relevant? It would be interesting if the authors performed statistical analysis with the Rockett controls to confirm the results.

e) According to the sample size even if the p-value was lower than 0.05 and due to multiple-testing, the statistical results indicated only suggestive associations. As the p-value is borderline, these findings need to be replicated in an independent sample set to be convincing.

f) In the Discussion, the authors indicated that no significant association was seen with *CR1* polymorphisms in the multi-centre candidate gene study (Rockett et al.), but the cases of the present study come from the Rockett's study and were compared to another control group. This information is confusing. It would be very important to confirm the association by using the Kenya controls from the Rockett's study to exclude potential selection bias in the control group. To perform this analysis, the authors should also take into account the complex interactions between *Sl2, McC^b^* and α^+^thalassaemia detected in the present study.

4) This reviewer's primary struggle with this paper is that it does not fully address how the known interaction between α-thalassemia and Sickle may be influencing or overlapping with these results. As the raw data are not included in the supplementary for these loci, the reader is forced to interpret these results with only the models and genotype frequencies that the authors have presented. This work uses substantially larger populations with better exposure controls than past reports on this topic, and thus after addressing several concerns, this work will be a valuable contribution to the community.

5) What is the population breakdown for the combined genotype that includes α-thalassemia, Sickle, *Sl*, and *Mc* together in each patient? Ideally this would also be broken down by disease outcome as well.

6) The binary parameterization of α-thalassemia described in Appendix 1 subsection “The *Sl2/Sl2* genotype is associated with protection against cerebral malaria in Mali” would appear to be miss-specified in light of Williams, 2005. It would seem more appropriate to model the double deletion vs. single or normal; or perhaps better in the early models to allow all three categories to compete for Sickle and Swain interactions. During model building the α-thalassemia-Sickle interaction may have been missed because of this, and it makes it difficult to determine whether the α-thalassemia-SL interaction is independent. Although the main effect term for Sickle is included in downstream adjusted models, there is not enough information to assess whether this factor would account for the α-thalassemia-Sickle interaction.

7) Cerebral malaria numbers are reported for the Mali cohort in Supplementary file 2, however associations with Knops are not discussed. It appears that these numbers may conflict with the trends in Kenya, and yet rosetting is still tested against these genotypes and presented as a potential mechanism of protection. Please reconcile this section or consider limiting the mechanistic connection to citing Rowe, et al.

---

## [Author Response]

Essential revisions:1) Given the dramatic reductions in severe malaria and deaths in the past decade, it is a bit strange that the studies were done with data collected during 2001-08 (case control study), and 1998-2001 (cohort study). So, a decade later, the authors have gone back to analyze stored samples. The implications of this are not clear and need to be clearly addressed. Are there time trends that can confound the associations? What about use of old samples and sample quality? Were the hypotheses pre-stated or generated post-hoc? (presumably the latter).

Making full use of existing datasets is of major importance in malaria genetic epidemiological studies, and there is no evidence to suggest that malaria phenotype/genotype associations vary over time. Historic datasets represent a huge investment of time and funding in establishing and collecting appropriate samples under logistically challenging conditions. Moreover, in many of the institutions with the greatest capacity for the ethical collection and processing of carefully phenotyped patient samples, the rates of severe malaria have declined in recent years. This is evident in some of the traditionally low-moderate malaria transmission areas of sub-Saharan Africa such as Kilifi in Kenya, the main focus of the current study. Unfortunately, most areas with continuing high malaria transmission and associated high mortality do not yet have the scientific infrastructure necessary to carry out large, high quality case-control studies. Hence, it is currently difficult to perform epidemiological studies on malaria using newly collected samples. We briefly describe these issues in the “Datasets studied” section of the Materials and methods which states:

“Historic datasets (i.e. >10 years old) are widely used in genetic epidemiological studies of malaria due to the logistical challenges of sample collection in malaria endemic countries and the changing epidemiological patterns of disease.”

All the samples used were of high quality and underwent stringent quality control. The “Sample processing and quality control for the Kenyan case-control study” section in the Materials and methods has been expanded to cover the quality control steps employed in sample collection and genotyping.

The hypotheses addressed in this study were generated after collection of the specimens, but prior to any exploratory analysis of the relevant datasets.

2) As regards the case-control study, I would need more convincing that cases and controls were derived from the same study base. The difference in age distribution will likely not matter for the genetic mutations, but what about implications for disease severity?

The cases and controls studied here were all derived from exactly the same area, which is the geographic region defined by the Kilifi Health and Demographic Surveillance System (KHDSS) (Scott et al., 2012). This is described in the Materials and methods, subsection “The Kenyan study area”, and a precise description of the cases and controls is described in an expanded section in the Materials and methods, subsection “The Kenyan case-control study”.

The controls were infant samples that were used to estimate the relevant genotype frequencies in the population from which the cases arose. This method has been widely used in African genetic association studies (e.g. Clarke et al., 2017; Busby et al., 2016; Band et al., 2013; Rockett et al., 2014) and is supported by general commentators (e.g. Rothman, 2014 and Vandenbroucke et al. International Journal of Epidemiology 2012). This is currently the most logistically feasible way of collecting sufficiently large numbers of control samples in many sub-Saharan African settings.

Text describing the rationale for using infant control samples has been added to the Materials and methods, subsection “The Kenyan case-control study”.

3) The authors indicated that the study included 5545 children from Kenya consisting in 1716 severe malaria cases and 3829 community controls.a) The authors performed the analysis on a subset of the samples used previously by Rockett et al. (2014). The authors should clarify why they did not use the total sample set of 2268 severe malaria cases included in Rockett et al. and how the 1716 samples were selected.

We apologise for the confusion caused by the lack of detail on how our study design differs from Rockett et al. The differences between our study and Rockett et al. is now fully explained in the “Comparison between this study and Rockett et al.” section in the Materials and methods (also Figure 3). See also the comments below.

b) As indicated by the authors, 407 individuals are excluded as lived out with KHDSS (Figure 3). Why? The authors should clarify this point. DNA not available? It would be surprising because all CR1 genotype (rs17047661 and rs17047660), sickle cell genotype, a^+^thalassaemia genotype and ABO blood group were available in the previous study of Rockett for the Kenyan population.

We excluded the severe malaria cases from outside the KHDSS because all of the controls came from within the KHDSS. Using only cases and controls from exactly the same area allowed the use of location as a random effect in the final statistical model, which greatly improved model fit. This is now described in the “Comparison between this study and Rockett et al.” section in the Materials and methods.

We have repeated our analysis including the severe cases of subjects who lived outside of the KHDSS study area (Supplementary file 1K), and this gives the same overall result as that shown in the main text. This is stated in the Materials and methods subsection “Comparison between this study and Rockett et al. (Rockett et al., 2014)”.

*c) It is not clear for me whether new genotyping has been performed for this study or if they re-used the genotypes obtained in the Rockett's study.*

We re-used genotypes from the Rockett’s study. This is now explained in the “Sample processing and quality control for the Kenyan case-control study” section of the Materials and methods, which states “The *Sl* and *McC* polymorphisms were originally typed as part of a larger study by Rockett et al., which included case-control data from 12 global sites (Rockett et al., 2014).”

The only additional genotyping of the Kenyan samples that was done for our study was for sickle cell trait and α ^+^thalassaemia, as described in the Materials and methods, subsection “Laboratory procedures”.

d) Why did the authors use controls recruited to an ongoing genetic cohort that were representative of the general population (Williams et al., 2009) as indicated in the subsection “The Kilifi case-control study” rather than used the controls of the previous study (Rockett et al.) that were also representative of the general populations? Is there a scientific reason? Are the Rockett controls not relevant? It would be interesting if the authors performed statistical analysis with the Rockett controls to confirm the results.

The controls in our study and the controls in Rockett et al. are the same. This is now explained in the Comparison between this study and Rockett et al.” section in the Materials and methods, which states:

“In both our study and Rockett et al. (Rockett et al., 2014), the control samples were identical and all came from within the KHDSS. Our study has 120 fewer controls than Rocket et al. (Rockett et al., 2014) due to missing genotypes, because we only used controls for whom full *Sl, McC,* sickle cell genotype,α^+^thalassaemia genotype and ABO blood group data were available.”

e) According to the sample size even if the p-value was lower than 0.05 and due to multiple-testing, the statistical results indicated only suggestive associations. As the p-value is borderline, these findings need to be replicated in an independent sample set to be convincing.

We agree that there is evidence, but not unequivocal evidence, for the suggested associations. The protective association between *Sl2* and cerebral malaria was first reported in a small case- control study from western Kenya (Thathy et al., 2005), but most subsequent studies have been underpowered. Our study is thus the first adequately powered independent sample set that replicates the protective association between *Sl2* and cerebral malaria. Furthermore, our study shows for the first time that *McC^b^* and α^+^thalassaemia influence the protective association of *Sl2* on cerebral malaria risk. Hence, our study contributes significantly to this debate and provides novel insights that may explain previous conflicting findings on CR1 polymorphisms and malaria.

We agree that further replication of our novel findings will be required. Unfortunately, because α^+^thalassaemia results from a deletion mutation, not a SNP, it cannot be detected on classical SNP-genotyping platforms, and has to be performed manually using labour intensive PCR-based methods (Chong et al., 2000), as in our study. The attempts of our collaborators in the MalariaGen consortium to impute α^+^thalassaemia genotypes from existing SNP genotyping data have not yet been successful. Hence, currently, to our knowledge, there are no other large sub-Saharan case-control studies that are typed for α^+^thalassaemia and *Sl* and *McC* polymorphisms that could be used for replication. Future work will address this need, but this is beyond the scope of the current manuscript, which is already complex, incorporating as it does results from two different epidemiological studies and functional data.

We have added text discussing the important issues of replication in the second and fourth paragraphs of the Discussion.

f) In the Discussion, the authors indicated that no significant association was seen with CR1 polymorphisms in the multi-centre candidate gene study (Rockett et al.), but the cases of the present study come from the Rockett's study and were compared to another control group. This information is confusing. It would be very important to confirm the association by using the Kenya controls from the Rockett's study to exclude potential selection bias in the control group. To perform this analysis, the authors should also take into account the complex interactions between Sl2, McC^b^ and α^+^thalassaemia detected in the present study.

As stated in response to 3d above, the controls used in our study are the same as the controls used in Rockett’s study.

As explained in response to 3a and 3b above, our study excludes some of the cases included by Rockett et al., due to their residence outside of the KHDSS. However, re-analysis including the additional cases studied by Rockett et al. (Supplementary file 1K) returns the same results as those shown in our main text. In other words, the difference in inclusion criteria does not explain why our results differ from Rockett.

The key distinctions that explain why we obtain a different result to Rockett et al. are:

i) We include *Sl* and *McC* in the same statistical model and adjust for confounding factors, including the statistically significant interaction with α^+^thalassaemia, whereas Rockett et al. looked at each SNP in isolation. This is now stated in the Materials and methods, subsection “Comparison between this study and Rockett et al. (Rockett et al., 2014)”, last paragraph.

The rationale for examining the effects of *Sl* and *McC* together in the same statistical model is that the two SNPs are only 33 bp apart in the CR1 gene, and each causes a change in amino acid charge (Sl: rs17047661 R1601G and McC: rs17047660 K1590E) with potential to impact upon the structure and function of the CR1 protein. Hence, it is biologically plausible that *Sl* might influence *McC* and vice versa.

ii) We are examining data from a single region in Kenya, whereas Rockett et al. analysed pooled data from 12 global sites. They state in their paper “…it is undoubtedly also the case that authentic genetic associations might be missed in multicentre studies if there is heterogeneity of effect across different study sites”.

Given that our data suggest that *McC^b^* and α^+^thalassaemia influence the protective association of *Sl2* with cerebral malaria, and that the frequencies of *McC^b^* (Figure 2) and α^+^thalassaemia are known to vary geographically, it would be predicted that the association of *Sl2* with cerebral malaria will vary by location, and may be missed by pooling data across different study sites.

We have summarised these arguments in the third and fourth paragraphs of the Discussion.

4) This reviewer's primary struggle with this paper is that it does not fully address how the known interaction between α-thalassemia and Sickle may be influencing or overlapping with these results. As the raw data are not included in the supplementary for these loci, the reader is forced to interpret these results with only the models and genotype frequencies that the authors have presented. This work uses substantially larger populations with better exposure controls than past reports on this topic, and thus after addressing several concerns, this work will be a valuable contribution to the community.

The potential effect of the α-thalassemia/sickle interaction on our results is answered in detail in a new section “Exploration of the negative epistasis between sickle trait and α^+^thalassaemia” in Appendix 2, and in new tables (Supplementary file 1N, 1O and 1P). See also comments below.

5) What is the population breakdown for the combined genotype that includes α-thalassemia, Sickle, Sl, and Mc together in each patient? Ideally this would also be broken down by disease outcome as well.

We have added data for the combined genotypes compromising sickle trait, α^+^thalassaemia and *Sl* for each clinical outcome in Supplementary file 1P.

6) The binary parameterization of α-thalassemia described in Appendix 1 subsection “The Sl2/Sl2 genotype is associated with protection against cerebral malaria in Mali” would appear to be miss-specified in light of Williams, 2005. It would seem more appropriate to model the double deletion vs. single or normal; or perhaps better in the early models to allow all three categories to compete for Sickle and Swain interactions. During model building the α-thalassemia-Sickle interaction may have been missed because of this, and it makes it difficult to determine whether the α-thalassemia-SL interaction is independent. Although the main effect term for Sickle is included in downstream adjusted models, there is not enough information to assess whether this factor would account for the α-thalassemia-Sickle interaction.

The parameterization of α^+^thalassaemia used in our model i.e. no α^+^thalassaemia alleles (normal) vs. one or more α^+^thalassaemia alleles was chosen in accordance with a previous report showing that both heterozygous (single deletion) and homozygous (double deletion) α^+^thalassaemia genotypes are associated with protection against severe malaria and death in the Kilifi area (Williams, Wambua, et al., 2005). A description of the parameterization of theα^+^thalassaemia variable can now be found in the Materials and methods, subsection “Statistical analysis”, second paragraph.

Investigation of interaction between sickle trait and α^+^thalassaemia is now described in the “Exploration of the negative epistasis between sickle trait and α^+^thalassaemia” section in Appendix 2. Analysis of our dataset confirms the existence of negative epistasis between sickle trait and α^+^thalassaemia in this population (Supplementary file 1N). However, of interest, this negative epistatic interaction was only seen in the severe malaria cases without cerebral malaria, whereas the α^+^thalassaemia/*Sl* interaction was specific to cerebral malaria cases. Therefore, the two interactions appear to be mutually exclusive. Therefore, we can confirm that for cerebral malaria in this dataset, no significant interaction exists between sickle trait and α^+^thalassaemia. This is the case no matter which α^+^thalassaemia parameterisation is used.

7) Cerebral malaria numbers are reported for the Mali cohort in Supplementary file 2, however associations with Knops are not discussed. It appears that these numbers may conflict with the trends in Kenya, and yet rosetting is still tested against these genotypes and presented as a potential mechanism of protection. Please reconcile this section or consider limiting the mechanistic connection to citing Rowe, et al.

There is no conflict between the Mali and Kenya data. A protective association between the *Sl2/Sl2* genotype and cerebral malaria was identified on analysis of the Mali dataset (aOR 0.35, 95% CI 0.12-0.89, p=0.024) and the *Sl2/Sl2-McC^a^/McC^a^* genotype combination was associated with protection against cerebral malaria (aOR 0.14, 95% CI 0.02-0.84, p=0.031). As such, we considered blood samples from this population to be appropriate for testing rosetting as a potential mechanism of action.

To make this clearer we have now included a sentence describing the protective association between *Sl2* and cerebral malaria in the Mali case-control to the Results, subsection “The *Sl2* allele was associated with reduced ex vivo rosette frequency in *P. falciparum* clinical isolates from Mali”.